# Lower novelty-related locus coeruleus function is associated with Aβ-related cognitive decline in clinically healthy individuals

Prokopis C. Prokopiou [1,9], Nina Engels-Domínguez [1,2,9], Kathryn V. Papp[3,4], Matthew R. Scott[4,5], Aaron P. Schultz[4,6], Christoph Schneider[1], Michelle E. Farrell[4], Rachel F. Buckley [3,4,7], Yakeel T. Quiroz [4,8], Georges El Fakhri[1], Dorene M. Rentz[3,4], Reisa A. Sperling[3,4], Keith A. Johnson[1,3,4] & Heidi I. L. Jacobs [1,2 ✉]

Animal and human imaging research reported that the presence of cortical Alzheimer's Disease's (AD) neuropathology, beta-amyloid and neurofibrillary tau, is associated with altered neuronal activity and circuitry failure, together facilitating clinical progression. The locus coeruleus (LC), one of the initial subcortical regions harboring pretangle hyperphosphorylated tau, has widespread connections to the cortex modulating cognition. Here we investigate whether LC's in-vivo neuronal activity and functional connectivity (FC) are associated with cognitive decline in conjunction with beta-amyloid. We combined functional MRI of a novel versus repeated face-name paradigm, beta-amyloid-PET and longitudinal cognitive data of 128 cognitively unimpaired older individuals. We show that LC activity and LC-FC with amygdala and hippocampus was higher during novelty. We also demonstrated that lower novelty-related LC activity and LC-FC with hippocampus and parahippocampus were associated with steeper beta-amyloid-related cognitive decline. Our results demonstrate the potential of LC's functional properties as a gauge to identify individuals at-risk for AD-related cognitive decline.

[1] Gordon Center for Medical Imaging, Department of Radiology, Massachusetts General Hospital, Harvard Medical School, Boston, MA, USA. [2] Faculty of Health, Medicine and Life Sciences, School for Mental Health and Neuscience, Alzheimer Centre Limburg, Maastricht University, Maastricht, The Netherlands. [3] Center for Alzheimer Research and Treatment, Department of Neurology, Brigham and Women's Hospital, Harvard Medical School, Boston, MA, USA. [4] Department of Neurology, Massachusetts General Hospital, Harvard Medical School, Boston, MA, USA. [5] Department of Biostatistics, Boston University, Boston, MA, USA. [6] The Athinoula A. Martinos Center for Biomedical Imaging, Department of Radiology, Massachusetts General Hospital, Harvard Medical School, Boston, MA, USA. [7] Melbourne School of Psychological Sciences, University of Melbourne, Parkville, VIC, Australia. [8] Department of Psychiatry, Massachusetts General Hospital, Harvard Medical School, Boston, MA, USA. [9] These authors contributed equally: Prokopis C. Prokopiou, Nina Engels-Domínguez. ✉email: hjacobs@mgh.harvard.edu

Alzheimer's disease (AD), the most prevailing type of dementia, is manifested by gradually progressive memory problems. The neuropathologic hallmarks include the deposition of beta-amyloid (Aβ) plaques and neurofibrillary tau tangles, which each emerge throughout the cortex in a distinct, predictable topographical manner, starting decades prior to clinical symptoms[1]. Several in-vitro and human autopsy studies suggested that one of the earliest sites in the brain implicated in these AD-related proteinopathies is the locus coeruleus (LC), a subcortical nucleus providing norepinephrine to the entire brain[2–4]. While many of these studies focused predominantly on the role of tau, LC neurons and their axonal terminals can accumulate soluble, oligomeric variants of Aβ early on, which interact with tau deposits in the LC and which subsequently trigger aggregation of soluble and extracellular Aβ into plaques in the remote cortex during the initial phases of AD[5].

As both the LC and Aβ undergo early pathologic alterations, there may be a common mechanism contributing to the initial clinical symptoms of AD. These initial symptoms emerge typically after age 60 when cortical fibrillar Aβ is detectable, tau is omnipresent in the LC and has reached the hippocampus (HIPP)[6]. Recent animal work demonstrated that oligomeric Aβ in the LC has the capacity to dysregulate LC activity promoting early hyperexcitability[7], which is reminiscent to the previously reported Aβ-driven excitatory toxicity in the cortical and hippocampal networks[8]. As the disease progresses and tau accrues, neuronal hyperexcitability was followed by neuronal silencing in transgenic mice[9,10] or loss of hyperconnectivity in fMRI studies[11].

Thus, a closer investigation of the effect of Aβ on activity of the LC and its functional connectivity (FC), in particular with the medial temporal lobe (MTL) could improve our understanding of the evolution and the early detection of AD-related cognitive decline and provide new anchor points for interventions.

LC neurons are known to discharge during conditions of novelty, arousal, and cognitive demand[12,13]. Animal studies have shown that novel or unexpected stimuli elicit phasic spikes in LC neurons, leading to NE release that targets the task-relevant regions in the brain, such as the amygdala (AMYG) and HIPP in the MTL and the prefrontal cortex[13–15]. Given that novelty detection is also an essential component of both learning and memory[14], tasks involving novelty detection may thus be well-suited to examine the vulnerability of the LC and the medial temporal pathways in AD-related cognitive decline. So far, few studies have looked at in vivo human LC function during novelty and its relationship with cognitive performance. A study by del Cerro et al.[16] demonstrated greater connectivity between the LC and the rest of the brain during oddball trials as compared to

standard trials, but the FC strength did not differ between healthy participants and mild cognitively impaired (MCI) patients. In contrast, Clewett et al.[17] used an emotional arousal paradigm in young adults and showed that increased LC-FC with the insula and dorsolateral prefrontal cortex was associated with subsequent better memory performance for negative stimuli. These studies provide initial evidence that communication of the LC with the cortex contributes to cognition, but to the best of our knowledge, no in vivo study has investigated the impact of AD pathology on LC activity or FC, and its downstream effects on cognitive decline.

In this work, we set out to investigate associations between in vivo LC activity and LC-MTL connectivity during the encoding of novel associations in well-characterized clinically unimpaired older individuals of the Harvard Aging Brain Study (HABS)[18]. We also examined whether novelty-related LC activity and connectivity have downstream effects on cognitive decline over a 10-year period as a function of Aβ using Aβ-PET. Based on the staging of pathology, suggesting proliferation of cortical tau in these older individuals[6], and its association with activity patterns, we hypothesized that novelty-related LC activity and connectivity would be lower at higher levels of Aβ, and that this lower novelty-related activity would be associated with Aβ-related cognitive decline.

## Results

**Characteristics of sample and design.** One hundred twenty-eight older individuals from HABS[18] underwent imaging, as well as longitudinal neuropsychological evaluations over up to 10 years. Seventy-one participants (55.46%) were female. At baseline, the mean age of the participants was 70.07 ± 8.86 (SD) and the mean education level was 15.74 ± 2.67 (SD) years. In addition, all participants had no history of medical or psychiatric disorders and were clinically unimpaired at baseline: Mini-Mental State Examination (MMSE)[19] >25, Clinical Dementia Rating (CDR)[20] = 0, and normal age- and education-adjusted scores on the Logical Memory delayed-recall test (Table 1). Furthermore, two different datasets were used in additional sensitivity analyses performed to demonstrate the robustness and reproducibility of our findings: (i) a dataset, referred to as Replication Dataset, consisting of forty-one individuals from HABS for replication purposes and, (ii) a dataset, referred to as Matched Dataset, consisting of 36 Aβ− and Aβ+ participants matched based on age, sex and years of education. The characteristics of these datasets are provided in (Supplementary Table 1) and (Supplementary Table 2) in the Supplementary material, respectively.

**Table 1 Characteristics of participants at baseline and follow-up time of neuropsychological evaluation sorted by PiB status.**

|  | Aβ− | Aβ+ | P value | |
|---|---|---|---|---|
| n, No. (%) | 92 (71.9) | 36 (28.1) | | |
| Age (years) | 66.75 [62.69, 72.69] | 76.62 [71.00, 82.38] | <0.001 | *** |
| Sex, No. (%) = M | 40 (43.5) | 17 (47.2) | 0.853 | |
| Education (years) | 16.00 [14.00, 18.00] | 16.00 [13.00, 18.00] | 0.946 | |
| PiB, PVC FLR (DVR) | 1.18 [1.14, 1.21] | 1.84 [1.50, 2.16] | <0.001 | *** |
| MMSE (score) | 30.00 [29.00, 30.00] | 29.00 [29.00, 30.00] | 0.051 | |
| GDS (score) | 2.00 [1.00, 4.00] | 2.00 [1.00, 4.00] | 0.889 | |
| Logical Memory delayed-recall (score) | 14.00 [12.00, 16.25] | 15.00 [12.00, 17.00] | 0.271 | |
| PACC5 (score) | 0.32 [−0.10, 0.57] | 0.03 [−0.32, 0.52] | 0.113 | |
| NP follow-up time (years) | 4.10 [2.03, 8.23] | 6.92 [3.92, 8.33] | 0.025 | * |

Data are presented as medians and [interquartile ranges (IQRs)] for continuous variables and proportions for dichotomous data. Two-tailed chi-square tests and Kruskal–Wallis tests were conducted to evaluate group differences.
DVR distribution volume ratio, FLR frontal, laterotemporal and retrosplenial cortices, GDS Geriatric Depression Scale, M male, MMSE Mini-Mental State Examination, NP neuropsychological evaluation, PVC partial volume corrected, PiB Pittsburgh Compound-B, PACC5 Preclinical Alzheimer Cognitive Composite.
*p < 0.05, ***p < 0.001.

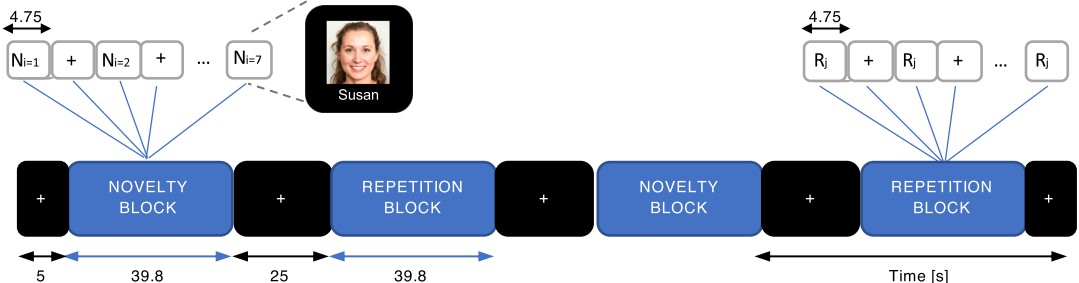

**Fig. 1 Design of fMRI-paradigm.** Diagram of the face-name associative paradigm. The task comprised events of unfamiliar and familiar face-name pairs organized within blocks of novelty and repetition, respectively. The novelty blocks consisted of 7 face-name pairs ($N_i$, $i = 1,..7$). The repetition blocks consisted of 7 trials during which two face-name pairs were alternated, one male and one female. ($R_j$, $j = 1,2$). The novelty, repetition and visual fixation (+) blocks, as well as the events within the blocks ($N_i$, $i = 1,...7$; $R_j$, $j = 1,2$; +) are depicted along with their corresponding duration. Each block was shown twice and alternated with visual fixation blocks. One functional run lasted for 4 min and 5 s, and a total of 6 functional runs were presented to each participant. The face shown in the diagram is fake, non-identifiable and was generated using artificial intelligence[91,92] for illustration purposes of the fMRI-task only.

The imaging assessments included both a Pittsburgh Compound-B (PiB) - positron emission tomography (PET) scan and a BOLD-fMRI session during which participants performed an encoding task of novel and repeated face-name associations, organized in blocks[21] (Fig. 1). The novelty block consisted of unfamiliar faces that varied in age, sex, and ethnicity paired with common first names. The repetition block consisted of repeated familiar faces, which were presented to the participants in a familiarization practice run prior to the fMRI session. Because of the responsiveness of the LC to novelty, here we focus on the novel versus repeated face-names contrast (NvR).

The LC is a remarkably tiny structure, located near multiple vessels and the fourth ventricle, thereby exposing the LC to motion and physiological noise[22,23]. To account for the confounding effect of non-neural related contributions in our measurements, the BOLD-fMRI images acquired during each condition were pre-processed, including AROMA (Automatic Removal of Motion Artifacts) for denoising and a custom ellipsoid smoothing kernel to account for the shape of the LC, and entered into a general linear model (GLM) for detecting task-related brain activation and generalized psychophysiological interaction analyses (gPPI) for detecting task-related voxel-wise functional connectivity of the LC within predefined regions of interest (ROIs; see "Methods"). In addition, to account for differences in hemodynamics across brain regions and individuals, we estimated region- and subject-specific hemodynamic response functions (HRF)[24]. We restricted our analyses to a set of predefined ROIs that are involved in memory, face processing, novelty detection and arousal: the AMYG, HIPP, parahippocampal gyrus (PHG), entorhinal (EC), temporal fusiform (TFC) insular (INS) cortices, and brainstem (medulla, pons, and midbrain; Supplementary Fig. 1c). As the LC was the seed for the gPPI analyses, we excluded the brainstem from the target ROIs. To demonstrate the robustness of our findings, we also performed several sensitivity analyses, which included analysis of unsmoothed data, time-series extracted from an eroded version of our LC ROI and correction for gray matter density, as well as analyses with the Replication Dataset and Matched Dataset (see "Methods").

**Distinct brain regions exhibit large individual variability in shape and amplitude of the BOLD hemodynamic response function.** We first examined the regional variability of the HRF obtained for each experimental condition (Novelty and Repetition) in the predefined set of ROIs. For the brainstem, we focused on the LC, which is involved in novelty detection and arousal[12]. The unknown HRFs were estimated directly from the data using a linear finite impulse response (FIR) model analysis, along with a function

expansion technique[25]. The overall shape of the estimated HRF for each condition and ROI was inspected using the first principal component of the HRF shapes at the group level (see "Methods").

For all ROIs the estimated HRF curve shape revealed similar characteristics for both experimental conditions (Fig. 2a). However, individual variability in both the shape and amplitude of the estimated HRF curves was observed in all ROIs. To illustrate this, representative HRF estimates of different participants obtained for the LC under both conditions are shown in Fig. 2b. In addition, the HRF amplitude (HRF peak value), which reflects the maximum instantaneous hemodynamic response to neuronal activity was significantly larger during Novelty as compared to Repetition in all ROIs, suggesting stronger neuronal activation during novel faces (Fig. 2c).

**Greater neuronal activity within predefined ROIs during Novelty versus Repetition.** First, we aimed to verify the brain activation patterns associated with novelty. We performed voxel-wise linear mixed-effects (LME) models with cross-sectional NvR contrast estimates as outcome variable, and including age and sex as covariates, random intercepts for participants and random slopes for fMRI runs (see "Methods"). Consistent with previous reports, we observed greater activation during NvR in the HIPP, and temporal occipital fusiform (TOF) cortices[21,26]. In addition, we also observed greater NvR activation in the AMYG, INS and LC (Fig. 3a). Our sensitivity analyses reproduced these observations when the unsmoothed data (Supplementary Fig. 2) and the Replication Dataset (Supplementary Fig. 3) were used, as well as when gray matter density was included into the model as a covariate (Supplementary Fig. 4). We detected no significant sex or age contributions in our activity maps. Given the age difference observed between the Aβ+ and Aβ− groups (Table 1) we post-hoc also repeated the same analysis using the Matched Dataset. The results (Supplementary Fig. 5) revealed greater activation during NvR in similar areas as the results obtained for the entire cohort shown in Fig. 3a. Adding PiB as covariate to the model did not modify the patterns of NvR activity. In addition, PiB did not interact with NvR on brain activation. Furthermore, greater activation was observed during Novelty compared to Fixation (NvF; Supplementary Fig. 6), albeit less than compared to NvR. No activation was observed during Repetition versus Fixation (RvF).

**Greater functional connectivity between locus coeruleus and amygdala as well as hippocampus during Novelty versus Repetition.** To investigate novelty-related coactivations between the LC and other voxels in the predefined set of ROIs, we employed gPPI analyses[27]. We performed LME models to

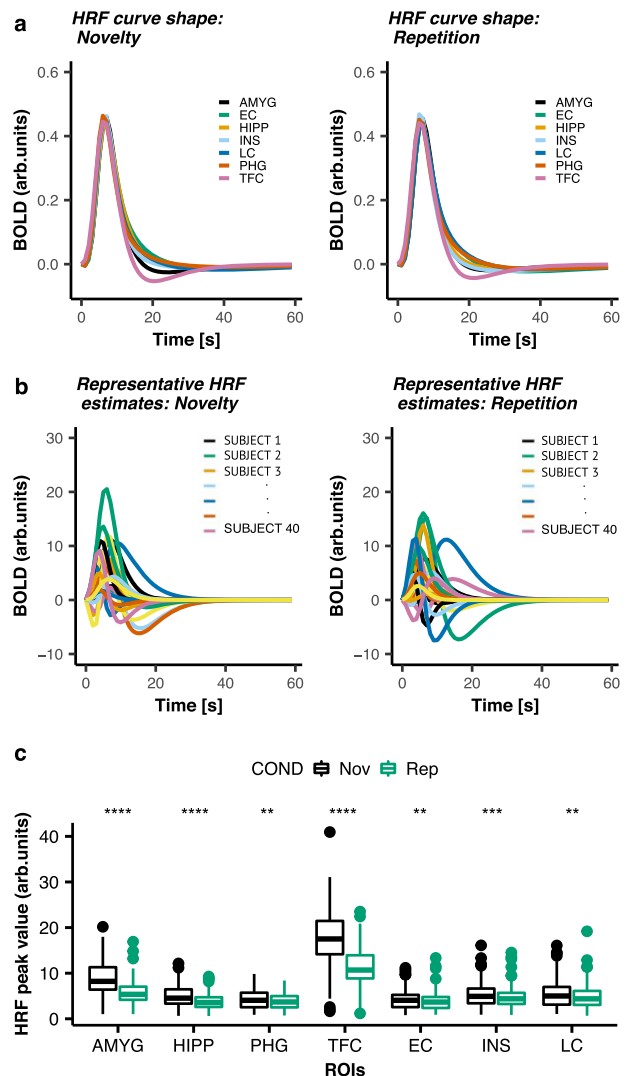

**Fig. 2 Regional variability of the hemodynamic response function.**
**a** Group HRF curve shapes obtained within ROIs during Novelty (left panel) or Repetition (right panel). **b** Representative subject-specific HRF estimates obtained for the LC during Novelty (left panel) and Repetition (right panel) for 40 representative participants (each color represents a different, randomly chosen participant). The x-axis is the time [seconds]. The y-axis is the BOLD signal intensity [arb. units]. **c** Statistical comparisons of the averaged HRF amplitude (HRF peak values) across runs obtained within ROIs during NvR (number of participants $n = 128$). On a group level the HRF curve shapes exhibited similar dynamics (peak latency and width (FWHM of the peak)) between the two experimental conditions. However, the HRF amplitude was significantly larger during Novelty compared to Repetition. ** $p < 0.01$; *** $p < 0.001$; **** $p < 0.0001$ (uncorrected); two-tailed paired t-test. Horizontal lines within boxes indicate the median. The bottom and top part of the boxes indicate the 25th and 75th percentile of the underlying HRF amplitude distribution, respectively. Dots represent outliers. Detailed statistics are provided in Supplementary Table 3 in the Supplementary material. Abbreviations: AMYG amygdala, COND condition, EC entorhinal cortex, HiPP hippocampus, INS insula, LC locus coeruleus, Nov novelty, PHG parahippocampal gyrus, Rep repetition, TFC temporal fusiform cortex.

identify brain regions whose coactivation with the LC differs between novelty and repetition, using their contrast estimates as outcome variable, and including age and sex as covariates, random intercepts for participants and random slopes for fMRI runs.

The resulting FC maps revealed greater NvR-related FC between the LC and both the bilateral AMYG and HIPP (Fig. 4a). Our sensitivity analyses reproduced these findings using BOLD time-series extracted from an eroded version of the LC ROI (Supplementary Figs. 7 and 8), unsmoothed data (Supplementary Fig. 9), as well as the Replication Dataset (Supplementary Fig. 10). We detected no significant sex or age contributions in the maps. Similar FC maps were obtained using the Matched Dataset (Supplementary Fig. 11) compared to the results obtained using our original dataset shown in Fig. 4a. Further, adding PiB as covariate to the model did not modify the patterns of NvR LC-FC and PiB did not interact with NvR on LC-FC. Also, greater FC was observed during both NvF and RvF between the LC and the bilateral AMYG and HIPP (Supplementary Fig. 12). However, FC during NvF was overall stronger and revealed more extended clusters of voxels within the AMYG and HIPP compared to RvF.

**Associations between novelty-related LC activity and functional connectivity with cognition.** Given that novelty processing promotes learning and memory[28,29], we sought to examine (i) the relationship between both NvR activity and FC between the LC and the respective individual voxels within the predefined set of ROIs (LC-FC), with cognitive performance, and (ii) whether these relationships are modulated by Aβ burden.

**Lower novelty-related activity in the LC is associated with Aβ-related PACC5 decline.** First, we examined which anatomic regions exhibited activity during NvR that is related to baseline PACC5 (Preclinical Alzheimer Cognitive Composite) performance using a voxel-wise linear model analysis including age, sex, and years of education as covariates. We observed no significant associations under cluster-extent thresholding (number of participants $n = 128$; cluster defining threshold $Z > 3.1$, two-tailed $p < 0.05$, family-wise error (FWER)-corrected). However, we also performed explorative analyses using false discovery rate (FDR)-based correction (number of participants $n = 128$; $Z > 2.3$, $P_{FDR} < 0.05$). This showed that lower NvR activation in small clusters of voxels in the right HIPP and left TFC (Supplementary Fig. 13) was associated with lower PACC5 performance.

Subsequently, we examined which anatomic regions exhibited activity during NvR that is associated with prospective PACC5 decline using voxel-wise mixed-effects model analyses including baseline age, sex and years of education as covariates (number of participants $n = 128$, number of observations is 753). Random intercepts and slopes were used for participants and time, respectively. No region was associated with PACC5 decline under cluster-extent-based thresholding. Using FDR correction we observed that lower NvR activation in the bilateral LC was associated with PACC5 decline (Supplementary Fig. 14; $Z > 2.3$, $P_{FDR} < 0.05$). Given that the PACC5 was developed as a sensitive measure of Aβ−related cognitive decline, we next aimed to uncover whether the associations between regions with NvR activity and PACC5 decline were modified by PiB.

To that end, we included the three-way interaction, NvR activity × time × PiB, in the linear mixed effect model. Using cluster-extent thresholding, we observed that lower NvR activation in the brainstem, including the right LC, and in the right PGH was associated with a steeper PACC5 decline, in particular when PiB was elevated (Fig. 5). The lateralization of these findings did not change under the less strict FDR correction (Supplementary Fig. 15). Similar results were obtained when PiB was used as a dichotomous rather than a continuous variable (Supplementary Fig. 16). To visualize these findings, we extracted the time-series from the right LC cluster (Fig. 5a) and plotted the simple slopes at mean and ±1 SD of LC activity for the two-way

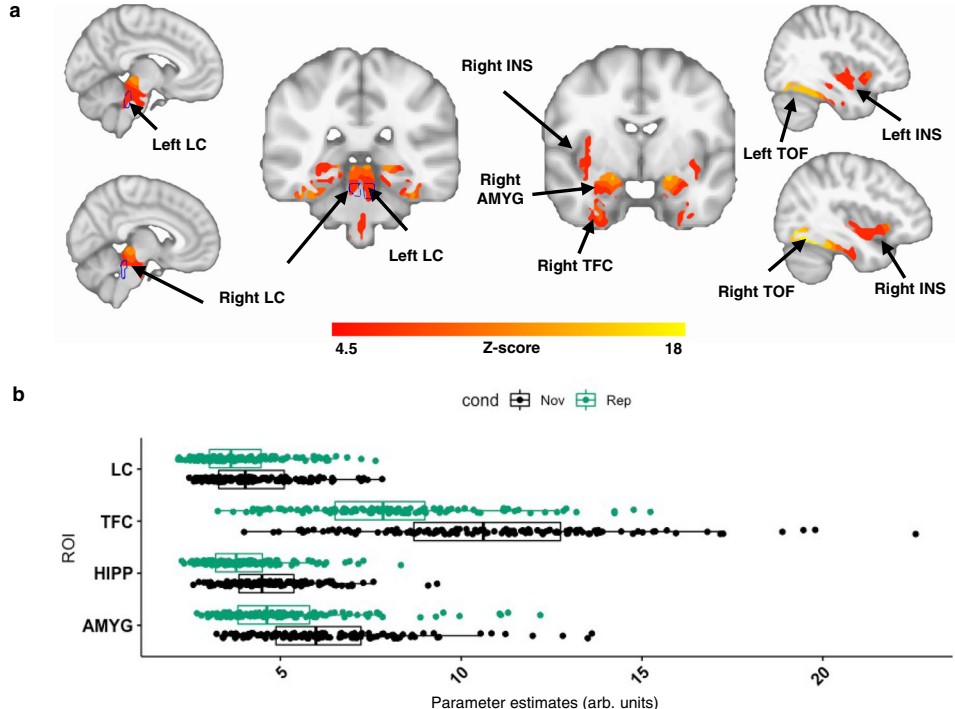

**Fig. 3 Voxel-wise analysis of brain activity in predefined regions of interest during Novelty versus Repetition. a** Brain activation maps obtained during NvR face-name stimuli: greater activation during NvR of voxels within the bilateral LC (indicated by the blue outline), amygdala, hippocampus, insula, temporal occipital fusiform cortices, entorhinal cortex and parahippocampal gyrus. Inference was performed using mixed-effects models including NvR contrast estimates as outcome variable, age and sex as fixed effects, random intercepts for participants, and slopes for fMRI runs. The brain activation maps were corrected for multiple comparisons using cluster-extent-based thresholding (number of participants $n = 128$; cluster defining threshold $Z > 4.5$, two-tailed $p < 0.05$, FWER-corrected). The outline of the LC ROI shown in blue was defined based on the Keren et al. (2009) LC atlas[84]. **b** Boxplots of averaged parameter estimate (PE) values across runs obtained for each participant and experimental condition in representative ROIs including the bilateral LC, temporal fusiform cortex, hippocampus and amygdala. Vertical lines within boxes indicate the median. The left and right part of the boxes indicate the 25th and 75th percentile of the underlying PE distribution, respectively. Dots represent averaged PE values across runs (number of participants $n = 128$). The ROIs were defined as the overlap between the regions exhibiting significant activation in (**a**) and anatomically defined regions derived from FreeSurfer (FS). The coordinates of the peak voxels of the detected clusters are provided in Supplementary Table 8. Abbreviations: AMYG amygdala, cond condition, HIPP hippocampus, INS insula, LC locus coeruleus, Nov novelty, Rep repetition, TFC temporal fusiform cortex, TOF temporal occipital fusiform cortex.

interaction (Fig. 5b, number of participants $n = 128$ and number of observations is 753; $B = 0.05$, $t(623) = 1.66$, $p = 0.097$, 95% confidence interval (CI)[−0.009, 0.11]) and the three-way interaction (Fig. 5c, number of participants $n = 128$ and number of observations is 753; $B = 0.20$, $t(621) = 3.37$, $p < 0.001$, 95% CI[0.09, 0.32]). Supplementary Fig. 17 illustrates the same result using dichotomous PiB. Post-hoc floodlight analyses revealed that the latter association is significant for PiB values equal to or above 1.44 DVR ($p < 0.05$ FDR corrected). We also investigated the Aβ-dependent associations between right PHG NvR activity and PACC5 decline (Supplementary Fig. 18) and observed that lower NvR PHG activity was associated with PACC5 decline when PiB values were equal to or above 1.62 DVR ($p < 0.05$ FDR corrected).

To examine whether the associations between LC NvR activity and (Aβ-related) PACC5 decline were driven by specific cognitive domains, we also investigated the subtests of the PACC5 and two other composite measures (Fig. 5d; Supplementary Tables 4 and 5). The largest effect sizes for the association between LC activation and Aβ-related cognitive decline were detected for the digit-symbol substitution and cognitive abilities test (verbal fluency).

**Lower novelty-related LC functional connectivity is associated with Aβ-related PACC5 decline**. To examine the relationship between LC-FC and PACC5, we took a similar approach as for the activation maps. We observed no significant associations

between LC-FC and baseline PACC5 performance when correcting for multiple comparisons using cluster-extent thresholding (number of participants $n = 128$; cluster defining threshold $Z > 3.1$, two-tailed $p < 0.05$, FWER-corrected). However, using FDR-based correction (number of participants $n = 128$; $Z > 2.3$, $P_{FDR} < 0.05$), we detected that lower FC between the LC and left AMYG, as well as left PHG are associated with lower PACC5 performance (Supplementary Fig. 19).

For the longitudinal data, we observed that lower novelty-related FC between the LC and left HIPP was associated with PACC5 decline (Fig. 6) at the cluster-extent thresholding levels. We then examined possible effect modification by PiB by including the three-way interaction in the LME model and observed that lower FC between the LC and bilateral HIPP and PHG are associated with steeper PACC5 decline in individuals with elevated PiB (number of participants $n = 128$ and number of observations is 753; cluster defining threshold $Z > 3.1$, two-tailed $p < 0.05$, FWER-corrected, Fig. 7a). Similar findings were observed when PiB was used as a dichotomous variable (Supplementary Fig. 20).

To visualize the associations between PACC5 decline and FC between the LC and areas in the MTL, we extracted the FC values from the voxels in the bilateral HIPP and PHG that survived the cluster-extent-based thresholding shown in Fig. 7a, and plotted the simple slopes at mean and ±1 SD of LC-FC for the two-way interaction (Fig. 7b, number of participants $n = 128$ and number

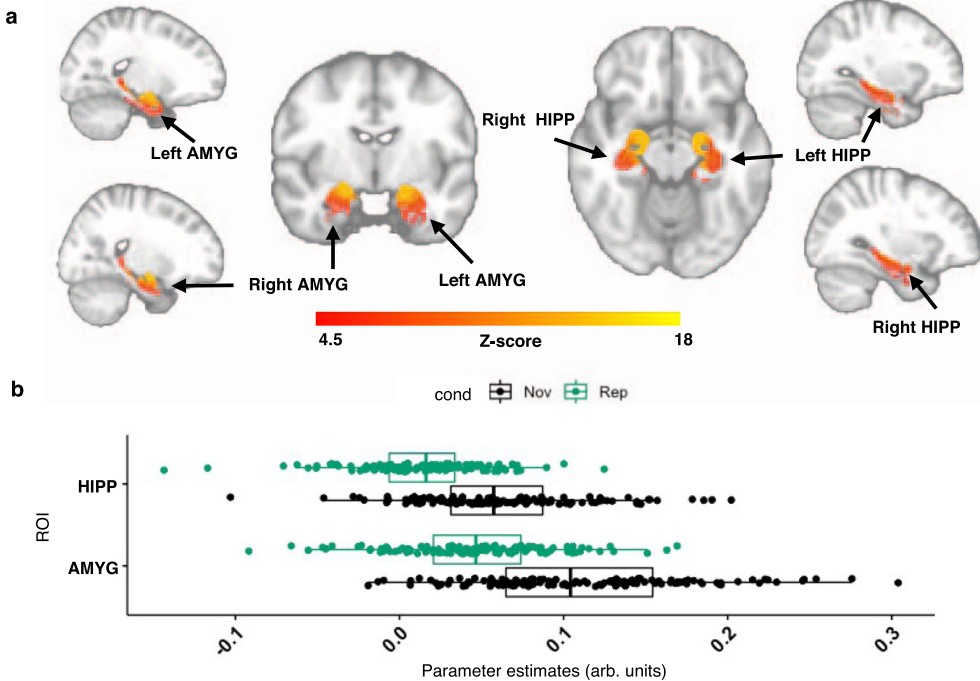

**Fig. 4 Voxel-wise analysis of LC functional connectivity in predefined regions of interest during Novelty versus Repetition. a** Functional connectivity maps between the LC and the predefined ROIs obtained during NvR face-name associations: greater FC between the LC and the amygdala as well as hippocampus during NvR. Inference was performed using mixed-effects models including NvR LC-FC contrast estimates as outcome variable, age and sex as fixed effects, random intercepts for participants and slopes for fMRI runs. The FC maps were corrected for multiple comparisons using cluster-extent-based thresholding (number of participants $n = 128$; cluster defining threshold $Z > 4.5$, two-tailed $p < 0.05$, FWER-corrected). **b** Boxplots of averaged gPPI parameter estimates (PE) for each participant and experimental condition in the bilateral hippocampus and amygdala. Vertical lines within boxes indicate the median. The left and right part of the boxes indicate the 25th and 75th percentile of the underlying PE distribution, respectively. Dots represent averaged PE values across runs (number of participants $n = 128$). The gPPI PE values obtained during Novelty were significantly larger than those during Repetition. The coordinates of the peak voxels of the detected clusters are provided in Supplementary Table 9. Abbreviations: AMYG amygdala, cond condition, HIPP hippocampus, Nov novelty, Rep repetition.

of observations is 753; $B = 1.24$, $t(623) = 3.42$, $p < 0.001$, 95% CI[0.53, 1.95]) and the three-way interaction (Fig. 7c; number of participants $n = 128$ and number of observations is 753; $B = 4.86$, $t(621) = 7.45$, $p < 0.001$, 95% CI[3.6, 6.13]). A post-hoc floodlight analyses revealed that this association is significant for PiB values above or equal to 1.25 DVR (p < 0.05 FDR corrected), which is below the GMM-derived PiB (PVC) cut-off value of 1.324 DVR in HABS. Supplementary Fig. 21b shows the three-way interaction for dichotomous PiB.

To examine the associations between LC-MTL FC and Aβ-related PACC5 decline across the various cognitive domains of the PACC5, we also investigated its subtests and two other composite measures (Fig. 7d). These analyses demonstrated significant associations between LC-MTL FC and Aβ-related cognitive decline for all the cognitive subtests of the PACC5 battery and the two additional composite measures (Supplementary Tables 6 and 7).

## Discussion

AD's neuropathologic hallmarks consist of Aβ plaques and neurofibrillary tau formations[1]. Neuronal hyperactivity has been linked to the emergence and progression of these proteinopathies, as well as to disease progression. Given the early involvement of the LC in AD's pathophysiology[2] and the fact that both animal[30,31] and human imaging studies[32] showed that the LC is highly responsive to novelty leading to NE-release in the HIPP and AMYG[13,33], thereby contributing to learning[14], we set out to investigate whether novelty-related LC activity and connectivity

are associated with cognitive decline over a 10-year period as a function of Aβ. To that end, we examined data from the well-characterized HABS cohort consisting of cross-sectional neuroimaging and longitudinal cognitive data.

Using dedicated processing methods to improve the measurement of the LC BOLD-fMRI signal and several sensitivity analyses confirming the robustness of our findings, we replicated animal work by showing that the LC shows higher activity during novelty and higher novelty-related FC with the bilateral AMYG and HIPP. We extended these results by demonstrating that both lower novelty-related LC activity and LC-MTL FC are associated with steeper Aβ-related cognitive decline. These findings are promising for the potential of LC's functional properties as a gauge to detect individuals at risk for AD-related processes. Interestingly, during the course of our study, ten participants progressed to MCI/AD, suggesting that our findings are also applicable to prodromal AD. Future studies with longer follow-up or a larger group of prodromal AD are needed to examine whether the relationship between LC function and Aβ-related cognitive decline varies as a function of disease stage.

Processing of novel stimuli is known to facilitate learning and memory[34,35]. Animal[30,31,36] and human pharmacological and imaging studies[37] demonstrated that exposure to novel stimuli induces activity of the LC, releasing NE, and leading to reconfiguration of specific task-relevant networks, such as the salience or memory-related networks. Consistent with these observations, during processing of novel faces, we observed higher activation in areas relevant for saliency, face discrimination, and learning. In addition, we observed higher FC between the LC and both the AMYG and HIPP

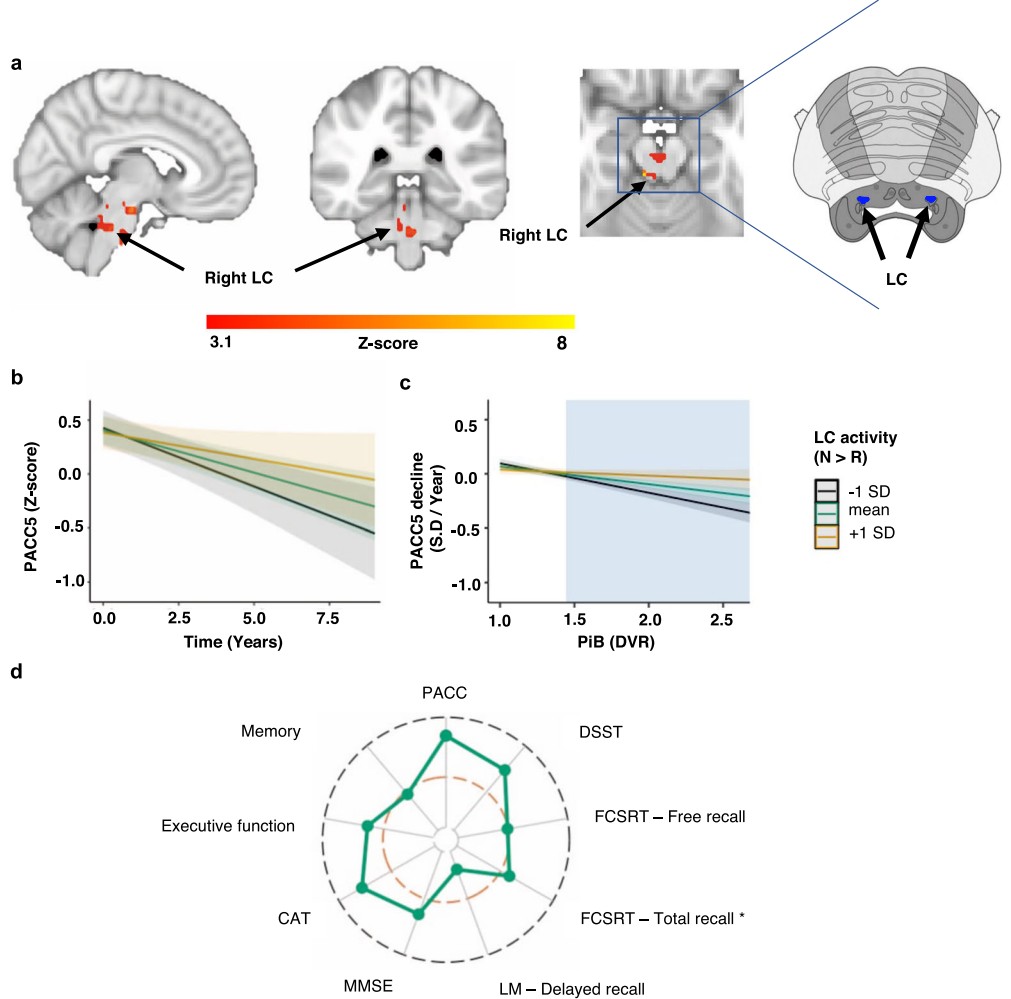

**Fig. 5 Lower novelty-related LC activity is associated with steeper Aβ-related PACC5 decline. a** Voxel-wise analyses relating NvR activity, PiB and longitudinal PACC5 measurements: lower NvR activation in the right LC and right parahippocampal gyrus are associated with greater decline on the PACC5 when PiB is elevated. Inference was performed using mixed-effects models including PACC5 as outcome variable, NvR contrast estimates, time, PiB, their interactions, age, sex and years of education as fixed effects, random intercepts for participants and slopes for time (number of years between baseline and follow-up cognitive assessments). Brain activation maps were corrected for multiple comparisons using cluster-extent-based thresholding (number of participants $n = 128$ and number of observations is 753; cluster defining threshold $Z > 3.1$, two-tailed $p < 0.05$, FWER-corrected). The insert on the right side shows the corresponding slice of the brainstem from the Duvernoy's atlas (adapted by permission from Springer Nature: Springer Science & Business Media)[93]. The LC is indicated with red markers. **b**, **c** Associations between PACC5 over time and LC activity during NvR. **b** Visualization of the two-way interaction between LC activity and time (number of participants $n = 128$ and number of observations is 753). **c** Visualization of the interaction between LC NvR activity and PiB on PACC5 slopes (number of participants $n = 128$). The cyan box illustrates the range of PiB values at which NvR LC activity is associated with PACC5 decline. In all line plots, the estimated marginal mean of the interaction terms is plotted at the mean (green), +1 SD (yellow) and −1 SD (black), but analyses were done continuously. Inference was performed using linear regression including PACC5 decline as outcome variable, and NvR LC activity, PiB, their interaction, age, sex, and years of education as predictor variables. Shaded areas around the fit lines show 95% CI. **d** Radar chart showing the magnitude of the associations (estimate/standard error) between LC NvR activity and PiB-related cognitive decline on the subtests of the PACC5, as well as the executive function and memory composite scores (number of participants $n = 128$ and number of observations is 753). The inner orange line indicated $t$-value = 1.96. The outer black line indicated $t$-value = 4.00. More detailed results are provided in Supplementary Table 5. * Random effects were modeled using only a random intercept for each subject. Abbreviations: CAT Category Fluency Test, DSST Digit-Symbol Substitution Test, DVR Distribution volume ratio, FCSRT Free and Cued Selective Reminder Test, LC locus coeruleus, LM Logical Memory, MMSE Mini-Mental State Examination, PiB Pittsburgh Compound-B, PACC5 Preclinical Alzheimer Cognitive Composite, SD standard deviation.

during novelty compared to repetition. Such a rearrangement for processing novel stimuli is thought to be important to reallocate cognitive resources for optimal performance, such as attentional shifts, learning, and action selection[33,38]. In addition, the involvement of the HIPP here aligns with the reported increase in hippocampal NE, which contributes to cellular mechanisms for effective learning, such as LTP[13,14,33].

Indeed, we observed that lower novelty-related bilateral LC activity and LC connectivity with the MTL were associated with

worse performance and steeper cognitive decline. This agrees with recent cross-sectional MRI studies suggesting that lower LC-cortical FC is also associated with worse memory performance for negative stimuli[17] and that age-related reductions in LC structural integrity are associated with impaired cognitive function and worse memory performance[39,40]. LC integrity has been also associated with tau pathology and longitudinal memory decline[41]. Further exploration of the PACC5 subtests revealed that these LC-network changes that we observed were predominantly related to performance on the

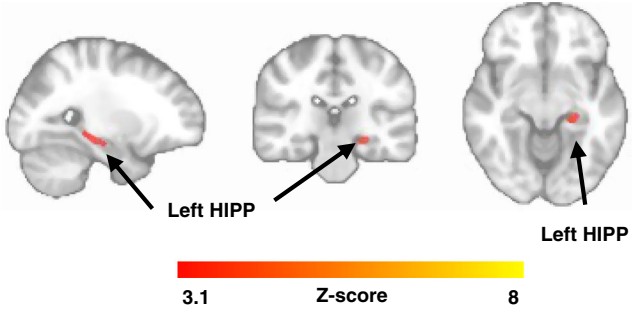

**Fig. 6 Lower novelty-related FC between the LC and the left hippocampus is associated with PACC5 decline.** Voxel-wise analysis relating NvR LC-FC and longitudinal PACC5 measurements: lower NvR FC between the LC and the left hippocampus is associated with decline on the PACC5. Inference was performed using mixed-effects models including PACC5 as outcome variable, NvR LC-FC contrast estimates, time, their interactions, age, sex and years of education as fixed effects, random intercepts for participants and slopes for time (number of years between baseline and follow-up cognitive assessments). The FC maps were corrected for multiple comparisons using cluster-extent-based thresholding (number of participants $n = 128$ and number of observations is 753; cluster defining threshold $Z > 3.1$, two-tailed $p < 0.05$, FWER-corrected). Abbreviation: HIPP hippocampus.

verbal fluency and digit-symbol substitution test (Supplementary Tables 4-7), tests requiring orienting and executive functions.

It is known that beyond the LC's effects on learning[22], its putative function involves fast disengagement from specific tasks and amplification of attentional focus to the goal-relevant information. Additionally, allocation of attention has generally been ascribed to the right hemisphere[42], which aligns with the abundant involvement of attention in the digit-symbol substitution test and our observation of lower right-sided LC activity being associated with steeper Aβ-related cognitive decline. This right lateralization is also consistent with the hypothesis that the LC may represent an important biological substrate underlying cognitive reserve and its associated processes, such as arousal, attention, and novelty[33,43], which all have been related to activation of predominantly the right frontoparietal network[44,45]. Previous imaging work showed lower connectivity between the LC and frontoparietal or salience networks resulting in greater distractibility in older individuals[46,47]. Our results indeed demonstrated that individuals who are able to maintain optimal levels of novelty-related LC functional properties may be more resilient to cognitive decline, even in the presence of elevated Aβ. Together, these findings strengthen the role of the LC-NE system in modulating networks, promoting cognition and potentially supporting cognitive reserve.

Even though novelty processing declines gradually during AD[1,48], we did not find an association between LC activity or LC-MTL FC and Aβ. Hyperactivity has been associated with elevated Aβ inducing an excitatory toxic environment for disease progression. It may be hypothesized that Aβ affects neuronal activity and networks differently depending on the disease stage. In fact, several studies using a similar task argue for a nonlinear process. In clinically unimpaired older individuals, hippocampal activity during a face-name associative memory task was not different between individuals with low or elevated Aβ, which is consistent with our observations for the LC[49,50]. However, Huijbers and colleagues reported modest differences in EC activity between low and elevated Aβ groups. In a study using the same fMRI task, hippocampal activity was higher in MCI individuals with elevated Aβ deposition, compared to those with low Aβ deposition[51]. But, in AD patients presented with this face-name associative

paradigm twice during six months, worse performance on the cognitive subscale of the Alzheimer's Disease Assessment Scale was associated with decreased novelty-related cortical activity[52].

We speculate that the impact of Aβ on activity or FC may be dependent on the targeted circuit. Previous imaging work reported combinations of hyper and hypoactivity in relationship to predominantly Aβ or tau pathology, respectively[53]. Work by Sepulcre et al.[53] highlights that there may be regional differences in vulnerability, possibly dependent on the topography of AD pathology. It should be noted that our sample consisted of older individuals (age range 50–89 years; $M = 70.07$, SD = 8.86), and at that time in life presence of pretangle material in the LC is ubiquitous. Unfortunately, we do not have information on the tau burden of our participants, but autopsy data suggests that it is very likely that the majority of our participants have Braak stage II pathology. In addition, given that 28% of our sample was classified with elevated Aβ, consistent with Thal phase II/III, it is presumed that these individuals bear at least Braak stage III-IV tau burden. Under this premise, we posit that the presence of tau may have obscured the impact of Aβ on neuronal activation. Animal work indicated that tau pathology, already when soluble non-aggregated, can dominate the effects of Aβ hyperactivity, resulting in suppression or even silencing of neuronal activity[10].

Consistent with a potential overriding effect of tau and the observation that tau is closely associated with cognition, we observed that lower novelty-related LC-MTL FC was associated with Aβ-related cognitive decline, at values below the GMM-derived threshold. Recent work demonstrated lower LC-cerebellar and LC-MTL FC patterns associated with reduced memory performance in offspring of patients with sporadic AD[54] and MCI patients with possible AD[55]. Our observations now indicate that lower novelty-related LC functional properties may identify clinically unimpaired individuals at risk of cognitive decline associated with an AD trajectory. Interestingly, we also note that individuals who are able to maintain optimal levels of novelty-related LC-activity or FC may be resilient to cognitive decline, even in the presence of elevated Aβ. Which LC-related factors may confer resiliency to AD pathology are not yet clear and warrant further examination. Animal research has suggested that greater novelty-related LC activity may be a potentially important component mediating the cognitive effects promoting cognitive reserve[56] via molecular mechanisms such as β-adrenergic enhanced neurogenesis[57] and elevated expression of plasticity-related genes[56,58].

This study has limitations. First, as the LC is one of the first regions affected by tau, it would have been interesting to examine the relationship between FC and cortical tau deposition. Unfortunately, tau-PET imaging was recently introduced in HABS, adding analytical complexities in terms of varying time difference with the fMRI data. Second, imaging the LC is challenging due to its proximity to the 4th ventricle and its tiny size, making it prone to partial volume effects. However, pairwise Pearson's correlation between BOLD-fMRI time-series extracted from the LC and 4th ventricle ROIs confirmed that our findings are not biased by partial volume effects (Supplementary Fig. 22). Similarly, we were not able to measure Aβ directly in the LC because of the low spatial resolution of PET imaging. Measuring LC function at 3T is challenging, due to the constraints on spatial resolution. To account for this, we implemented brainstem targeted pre-processing techniques including (i) weighted registration of the brainstem into the Montreal Neurological Institute (MNI)-152 space, (ii) nuisance regression including the average BOLD time series of the 4th ventricle, and (iii) special smoothing using an ellipsoid Gaussian kernel aiming to improve the spatial SNR and enhance detection of elongated structures within the brainstem, such as the LC. This ellipsoid smoothing brought the resolution of our data to a comparable and sometimes even better

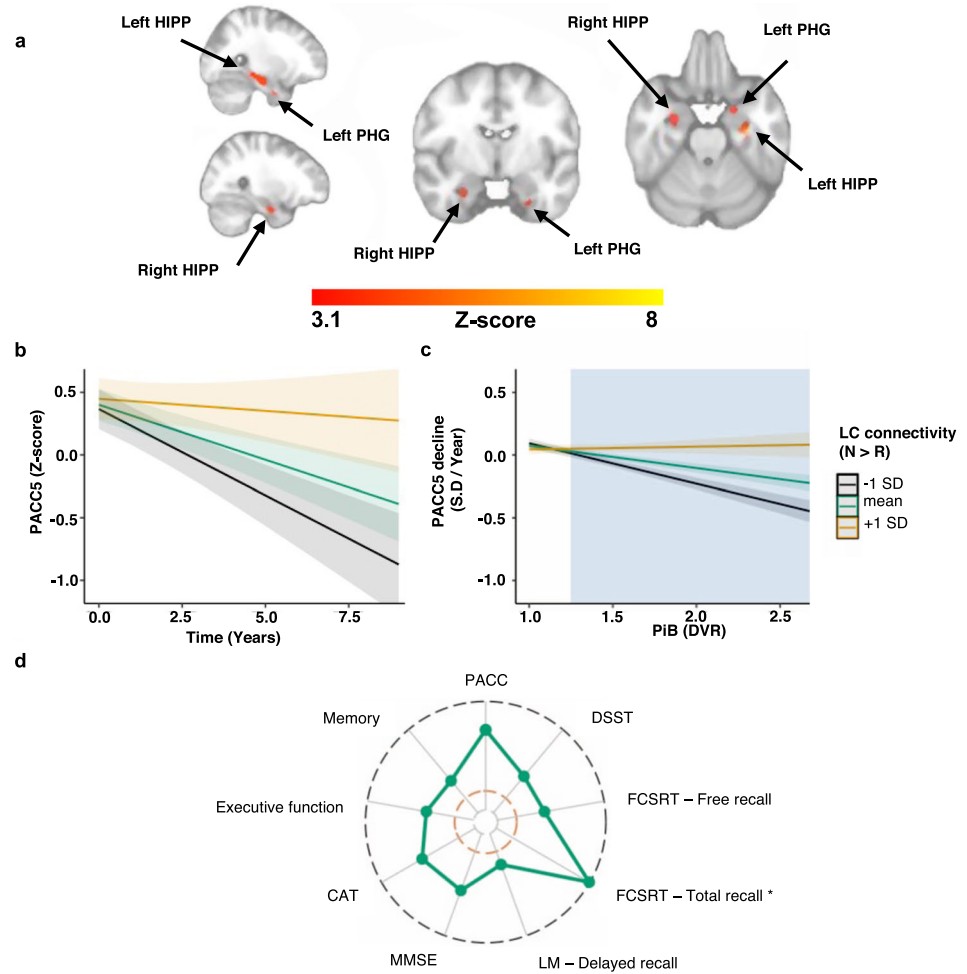

**Fig. 7 Lower novelty-related FC between the LC and bilateral hippocampus as well as parahippocampal gyrus are associated with steeper Aβ-related PACC5 decline. a** Voxel-wise analyses relating LC- region of interest FC, PiB, and longitudinal PACC5 measurements: lower NvR functional connectivity between the LC and the bilateral hippocampus and parahippocampal gyrus is associated with greater decline on the PACC5, in particular in individuals with elevated PiB. Inference was performed using mixed-effects models including PACC5 as outcome variable, NvR LC-FC contrast estimates, time, PiB, their interactions, age, sex, and years of education as fixed effects, random intercepts for participants, and slopes for time (number of years between baseline and follow-up cognitive assessments). The maps were corrected for multiple comparisons using cluster-extent-based thresholding (number of participants $n = 128$ and number of observations is 753; cluster defining threshold $Z > 3.1$, two-tailed $p < 0.05$, FWER-corrected). **b** Visualization of the association between PACC5 performance over time and NvR FC between the LC and the group of voxels within the bilateral hippocampus and parahippocampal gyrus shown in Fig. 7a. **c** Visualization of the interaction between NvR LC-FC and PiB on PACC5 slopes (number of participants $n = 128$). The cyan box illustrates the range of PiB values at which lower NvR LC- hippocampus and parahippocampus FC is associated with PACC5 decline. In all line plots, the estimated marginal mean of the interaction terms is plotted at the mean (green), $+1\,SD$ (yellow), and $-1\,SD$ (black), but analyses were done continuously. Inference was performed using linear regression including PACC5 decline as outcome variable, and NvR LC-FC, PiB, their interaction, age, sex and years of education as predictor variables. Shaded areas around the fit lines show 95% CI. **d** Radar chart showing the magnitude of the associations (estimate/standard error) between NvR LC- hippocampus and parahippocampus FC and PiB-related cognitive decline on the subtests of the PACC5, as well as the executive function and memory composite scores (number of participants $n = 128$ and number of observations is 753). The inner orange line indicates $t$-value $= 1.96$. The outer black line indicates $t$-value $= 10.00$. More detailed results are provided in Supplementary Table 7. * Random effects were modeled using only a random intercept for each subject. Abbreviations: CAT Category Fluency Test, DSST Digit-Symbol Substitution Test, DVR Distribution volume ratio, FCSRT Free and Cued Selective Reminder Test, HIPP hippocampus, LC locus coeruleus, LM Logical Memory, MMSE Mini-Mental State Examination, PHG parahippocampal gyrus, PiB Pittsburgh Compound-B, PACC5 Preclinical Alzheimer Cognitive Composite and SD Standard Deviation.

resolution than other LC studies using spherical smoothing (Supplementary results 1). Furthermore, our sensitivity analyses performed using unsmoothed data (providing the highest spatial resolution), an eroded version of the LC ROI, and the Replication and Matched Datasets demonstrated the robustness and reproducibility of our imaging findings. Finding a replication dataset large enough that would allow us to perform similar sensitivity analyses for the association of LC function with Aβ-related cognitive decline is challenging and remains to be performed in a future study. Finally, the proximity of the LC to the 4th ventricle

makes it susceptible to physiology artifacts, such as cardiac pulsatility and respiration. Proper physiology noise removal would require acquisition of physiology measurands. In the absence of these, we incorporated additional nuisance regression using the average white matter and lateral ventricle CSF signal, as well as ICA (Independent Component Analysis)-based blind source separation techniques (ICA-AROMA (Automatic Removal of Motion Artifacts)[59]) aiming to separate the sources of physiological and motion noise from the signal of interest as much as possible, and hence improving temporal SNR of our data.

In conclusion, we investigated the association between in vivo novelty-related LC activity or LC-MTL functional connectivity and Aβ or Aβ-related cognitive decline in clinically unimpaired older individuals. Our findings demonstrate that lower novelty-related LC activity and functional connectivity between the LC and HIPP and PHG were associated with steeper decline in cognition measured over 10 years, particularly in the presence of elevated Aβ deposition. These results emphasize the potential of functional properties of the LC as a gauge to differentiate individuals vulnerable for the AD-related trajectory from those carrying resilience against AD-related changes.

## Methods

**Participants.** One hundred twenty-eight individuals from HABS[18] were included in the present study (57 M/71 F; median age at baseline = 69.62, interquartile range (IQR): 63.69–76.81 years; median years of education at baseline = 16, IQR: 14–18 years; Table 1). HABS is an ongoing longitudinal observational study of cognitively unimpaired individuals aimed to further our understanding of normative aging and preclinical AD. All participants underwent baseline task-fMRI and PiB-PET imaging (within one year of the MRI scan), and annual cognitive assessments (followed up for up to 10 years; median years = 4.25, IQR: 2.07–8.26 years). For the cognitive assessments, $n = 125$ participants completed 2 visits, $n = 117$ participants completed 3 visits, $n = 89$ participants completed 4 visits, $n = 78$ participants completed 5 visits, $n = 60$ participants completed 6 visits, $n = 55$ participants completed 7 visits, $n = 50$ participants completed 8 visits, $n = 43$ participants completed 9 visits and $n = 8$ participants completed 10 visits (Supplementary Fig. 23). To validate our imaging results, we analyzed two different fMRI datasets. The first one, the Replication Dataset, consisted of fMRI data acquired from forty-one older individuals using an alternative version of the face-name associative task. Twenty-four individuals overlapped with the main cohort but were scanned four years later using an alternative version of the face-name associative task. The other seventeen participants joined HABS later in the study and their baseline imaging and cognitive measurements were not within one year from each other and were therefore excluded from the main sample. The characteristics of the Replication Dataset are provided in Supplementary Table 1. The other dataset, the Matched Dataset, consisted of a subset of 36 Aβ- individuals who were matched to 36 Aβ+ individuals based on the age, sex, and years of education distributions using propensity-based matching. The characteristics of the Matched Dataset are provided in Supplementary Table 2. The study complied with all ethical regulations and was approved by the Partners Human Research Committee at Massachusetts General Hospital. All participants provided written informed consent and received monetary compensation after each visit. Participants had no history of medical or psychiatric disorders and were clinically unimpaired at baseline: median MMSE[19] = 29 (IQR: 29–30), CDR[20] = 0, and normal age- and education- adjusted scores on the Logical Memory delayed-recall test. History of alcoholism, drug abuse, head trauma, or a family history of autosomal dominant AD were additional exclusion criteria.

Clinical disease progression during the course of the study was determined based on a consensus diagnosis of MCI and AD dementia. The criterion for MCI diagnosis was based on a (i) global CDR of 0.5 or (ii) performance on any domain-specific composite score (memory, processing speed, and executive function) lower than 1.5 standard deviations below the sample mean[60]. The CDR was completed by accredited neuropsychologists and psychiatrists. All CDR raters were independent and blinded to participant biomarker status. Participants meeting these criteria were brought to a consensus meeting conducted by a multidisciplinary team of clinicians. Diagnosis was based on clinical consensus after reviewing multiple consecutive CDR evaluations, cognitive data including the Logical Memory delayed-recall score, MMSE and Geriatric Depression Scale (GDS)[61], as well as relevant medical history. Based on this evaluation, 10 participants of our main sample progressed to MCI, and 5 of them progressed further to AD dementia over the 10-year period of the study.

**Experimental design.** Participants were asked to perform a face-name associative memory paradigm that instructs participants to learn face-name pairs[21]. The stimuli consisted of unfamiliar faces that were shown either a single time as a novel face or multiple times as a repeated face. The faces varied in age, sex and ethnicity, and were presented in color against a black background. The name was presented in white letters beneath. The stimuli were organized in blocks of novelty and repetition events, respectively (Fig. 1). The novelty blocks consisted of 7 face-name pairs. The repetition blocks consisted of 2 face-name pairs, one male and one female, which were first shown to the participants in a familiarization practice run prior to the fMRI runs. The face-name pairs within each block were presented for 4.75 s and followed by a brief, randomly jittered white fixation crosshair, giving a total duration of approximately 40 s per block. Participants were asked to remember the name associated with each face, and to indicate with a button press whether or not they thought the name was a good match for the face. The latter was a purely subjective decision of each individual that was used to ensure engagement with the task[26]. The task comprised 6 functional runs, and each type of block was shown twice in each run in an interleaved fashion. In addition, visual fixation blocks were added: a 5-s fixation block was presented at the beginning and end of each run and a 25-s fixation block was alternated with the novelty and repetition blocks. We focused primarily on the novelty versus repetition contrast.

**Cognitive performance.** We employed the most recent version of the Preclinical Alzheimer's Cognitive Composite score (PACC5)[62,63]. PACC5 is designed to track early Aβ-related cognitive decline. The PACC5 score consists of the average of the z-transformed (using baseline mean and standard deviation) scores of the Digit-Symbol Substitution Test (DSST)[64]; free and total recall elements of Free and Cued Selective Reminding Test (FCSRT)[65]; Logical Memory Delayed Story Recall[66]; Mini-Mental State Examination (MMSE)[19]; and Category fluency (CAT) tests to animals, fruits, and vegetables[67]. We allowed at most one missing subtest for PACC5 score calculation. Missing subtests were excluded from the calculation. In addition to PACC5, we also included a memory and an executive function composite score following factor analyses[68]. The memory composite score comprised z-transformations of the delayed recall scores of the 6-Trial Selective Reminding test[69], free recall element of the FCSRT[65] and Logical Memory Delayed Story Recall[66]. The executive function composite score comprised z-transformations of the Trail Making Test form B – A[70], Letter Number Sequencing test[71] and phonemic fluency FAS test[72]. The intraclass correlation coefficient (ICC)[73] demonstrated adequate measurement reliability over time. For the PACC5, the ICC was 0.85, for the memory composite score 0.81, and for the executive function composite score 0.81.

**PET data acquisition and pre-processing.** PiB-PET data were acquired on a Siemens ECAT EXACT HR + PET system located at the Massachusetts General Hospital (3D mode; 63 image planes; 15.2 cm axial field of view; 5.6 mm transaxial resolution; and 2.4 mm slice interval). Following radiosynthesis of $^{11}$C PiB[74], an injection of 8.5-15 mCi PiB was administered and dynamic data in 69 frames was obtained for 60 min (12 frames × 15 s, 57 frames × 60 s). The $^{11}$C PiB-PET data were registered to the subject's high-resolution anatomical MRI and converted into distribution volume ratio (DVR) using the Logan graphical method[75].

Cerebellar gray matter was used as the reference region[76]. The PET images were aligned to the high-resolution anatomical MRI images described below, and motion corrected using cross-modal alignment in SPM12 (Welcome Department of Cognitive Neurology, Function Imaging Laboratory, UK). Partial volume correction (PVC) was performed using the Geometrical Transfer Matrix approach implemented in FS. PiB retention was assessed using a large ROI comprising the frontal, lateral temporal and retrosplenial (FLR) cortices[77,78]. Classification into Aβ+/− groups was ascertained using a PiB cut-off value of 1.324 for the PVC data (1.20 for non-PVC), which was previously determined with a Gaussian mixture modeling approach on the entire HABS cohort[18]. Based on this cut-off value, 92 participants in this study were classified as low Aβ (Aβ-), and 36 participants (28.13% of the entire cohort) with elevated Aβ (Aβ+) at baseline. This is consistent with the estimated prevalence of Aβ+ individuals among cognitive unimpaired individuals ranging from 13% to 30%[79]. The median baseline PiB-PET-imaging delay from the first neuropsychological evaluation was 0.29 years or 107 days (IQR, 0.20–0.46 years) and the median delay from the first MRI scan was 0.09 years or 35 days (IQR, 0.04-0.19 years).

**MRI data acquisition and pre-processing.** All data were collected on a 3T Trio-Tim syngo MR B17 scanner (Siemens Medical Systems) using 12-channel phased-array head coils for reception located at the Athinoula A. Martinos Center for biomedical imaging in Charlestown, MA. Functional data were acquired using a T2*-weighted echo planar imaging (EPI) sequence sensitive to BOLD contrast. Sequence parameters: TR/TE (Repetition/Echo Time) = 2000/30 ms; Voxel size = 3.1 × 3.1 × 5.0; FA (Flip Angle) = 90°; 30 slices; Acquisition matrix = 200 × 200 × 179 mm. Each run comprised 127 volumes with slices acquired in an interleaved manner in a coronal orientation perpendicular to the anterior commissure-posterior commissure (AC-PC) line. This orientation was chosen to maximize the in-plane resolution within the HIPP and brainstem[21]. In addition, a high-resolution T1-weighted Magnetization Prepared Rapid Acquisition Gradient Echo structural image was also acquired to aid registration of the PiB-PET and BOLD images to a common MNI space. Sequence parameters: TR/TE = 2300/2.95 ms; Voxel size = 1.1 × 1.1 × 1.2 mm; Inversion time = 900 ms; FA = 9°; 176 (sagittal oriented) slices; Acquisition matrix = 270 × 254 × 212 mm; 2× (GRAPPA) acceleration.

The high-resolution T1 structural images were pre-processed in FS (version 6.0; http://freesurfer.net) using the software package's default automated reconstruction protocol, which included: conversion to 1 mm³ voxel size, motion correction, transformation to Talairach space, intensity normalization, skull stripping, segregation of left and right hemispheres, brainstem and cerebellum removal, correction of topology defects, detection of gray matter/white matter and gray matter/CSF borders, and parcellation of cortical and subcortical areas. In a separate process, the brainstem was parcellated into the medulla oblongata, pons, midbrain and superior cerebellar peduncle using a Bayesian segmentation algorithm implemented in FS[80].

The basic pre-processing of the BOLD images was carried out using the Oxford Centre for Functional Magnetic Resonance Imaging of the Brain Software Library (FMRIB, UK – FSL version 5.0.7)[81]. This included brain extraction, slice timing correction, motion correction via volume realignment, normalization to the 2 mm$^3$ MNI-152 EPI template, and further artifact detection and removal using ICA-AROMA[59], followed by detection and removal of motion-contaminated volumes (i.e., scrubbing) based upon the derivative of root mean squared variance over voxels (DVARS)[82]. To improve the accuracy of the registration of the brainstem, we initially aligned the BOLD images to the high-resolution 1 mm$^3$ - T1 structural image of each subject using boundary-based registration[83]. Subsequently, the T1 structural image was aligned with the MNI-152 template using a 3-step registration procedure: in the first step the T1 structural image was registered to the MNI-152 template using an affine, linear registration with 12 degrees of freedom. In the second step, this affine registration was refined using cost-function weighting input and reference volumes (Supplementary Fig. 24). The weighting volumes were constructed by assigning higher weights to the voxels within the 4$^{th}$ ventricle, midbrain, pons and medulla. In the third step, a nonlinear registration was performed, which was initialized using the transformation matrix obtained from the previous step. To mitigate artifacts from CSF flow, breathing motion and cardiac pulsatility of blood vessels in the brainstem[23], nuisance regressors were generated and removed from the data through linear regression. The nuisance regressors included three ROI time-series obtain as the mean across voxels in the 4th ventricle, lateral ventricles and white matter, the 6 motion parameters (MPs) generated during volume realignment, the derivatives of the 6 MPs, and the squares of all the aforementioned time-series. Finally, the BOLD images were spatially filtered using a custom ellipsoid Gaussian kernel (Supplementary Fig. 24) in order to enhance detection of elongated structures within the brainstem, such as the LC. To demonstrate spatial smoothing did not bias our results by introducing signal from the CSF, we performed sensitivity analyses for NvR brain activation or LC-MTL FC during NvR using unsmoothed data (Supplementary Fig. 2, Supplementary Fig 9).

A predefined set of ROIs was included for voxel-wise analyses of brain activity and functional connectivity of the LC during Novelty versus Repetition (Supplementary Fig. 1c). These regions were chosen based on their involvement in novelty and arousal, learning or face processing, and excluding regions that may suffer signal loss in coronal acquisitions. The selected ROIs included the AMYG, EC, HIPP, INS, TFC, and brainstem. All the ROIs except the LC were defined at the subject level using the subject-specific parcellation obtained with FS.

In addition, we created an LC template as anatomical reference guide: first, we registered an existing postmortem validated template of the LC[84] to the individual T1 image of each subject using the diffeomorphic transformation of the Advanced Normalization Tools (ANTs)[85]. Then, we constructed a template of the functional space of the subjects' BOLD-fMRI data using the ANTs multivariate template construction tool (version 2.1.0; http://stnava.github.io/ANTs/). Next, we registered the image of LC template from native structural space of each subject into the template functional space and constructed a map of high group consensus of the LC across participants; that is, a region where the greatest group convergence of the LC is observed (Supplementary Fig. 8—red ROI). In addition, an eroded version of the LC template was manually constructed (60% volume reduction) for sensitivity analyses (Supplementary Fig. 8—blue ROI). Lastly, this group map of the LC was warped to the native functional space of each subject using the FSL's linear imaging registration tool.

**fMRI data analysis.** The BOLD-fMRI data were analyzed in MATLAB R2018b (https://www.mathworks.com) using a system-theoretic approach. In this context, the BOLD signal was modeled as the sum of the output of two parallel block-cascade linear, FIR models with each block corresponding to a different experimental condition as follows:

$$y(n) = \underbrace{\sum_{m_1=0}^{M} h_1(n-m_1)x_1(m_1)}_{\text{condition 1}} + \underbrace{\sum_{m_2=0}^{M} h_2(n-m_2)x_2(m_2)}_{\text{condition 2}} + e(n), \quad (1)$$

where $y(n)$ denotes the system output (i.e. BOLD signal change), $x_1(n)$ the time-series of the first condition (i.e. novelty), $x_2(n)$ the time-series of the second condition (i.e. repetition), $h_1(n)$ the HRF associated with the first condition, $h_2(n)$ the HRF associated with the second condition, and $e(n)$ additive noise.

The HRF, which reflects the BOLD signal change in response to a neural event associated with an experimental condition (e.g. seeing a new face-name pair), was estimated efficiently from the data using function expansions in terms of orthonormal bases[25] given by:

$$h_i(n) = \sum_{j_i=1}^{L_i} c_{i,j_i} b_{i,j_i}(n), \quad (2)$$

where $i=1,2$ indicates the experimental condition, $B(L_i,M) = \{b_{i,j_i}(n) = 1, \dots, L_i; n = 1, \dots, M\}$, a set of $L_i$ orthonormal basis functions, $M$ the finite system memory, and $c_{i,j_i}$ the unknown expansion coefficients. Substituting (2) into

(1), the model for the BOLD signal can be re-expressed as:

$$y(n) = \sum_{j_1=1}^{L_1} \sum_{m_1=0}^{M} c_{1,j_1} b_{1,j_1}(n-m_1)x_1(m_1) + \sum_{j_2=1}^{L_2} \sum_{m_2=0}^{M} c_{2,j_2} b_{2,j_2}(n-m_2)x_2(m_2) + e(n),$$
$$= \sum_{j_1=1}^{L_1} c_{1,j_1} v_{1,j_1}(n) + \sum_{j_2=1}^{L_2} c_{2,j_2} v_{2,j_2}(n) + e(n), \quad (3)$$

where $v_{i,j_i}$ denotes the convolution of the $i$th condition time-series with the $j_i$-th basis function. Equation (3) can be re-written in a compact matrix form as:

$$\mathbf{Y} = \mathbf{Vc} + \mathbf{e}, \quad (4)$$

whereby the unknown expansion coefficients $\mathbf{c}$ can be estimated directly from the data using ordinary least squares.

The selection of a basis set depends on the dynamic behavior of the system to be modeled and a proper selection may yield more parsimonious BOLD signal representations. A popular selection for modeling physiological systems is the Laguerre basis[86]. The Laguerre basis functions are orthonormal in $[0,\infty)$ and exhibit exponentially decaying behavior that make them suitable for modeling causal systems with finite memory. In this work, we employed a basis set that is based on a smoother variant of the Laguerre basis, the spherical Laguerre basis[87], which allows robust HRF estimation from BOLD-fMRI measurements. The j-th spherical Laguerre function is given by:

$$d_j(n) = \sqrt{\frac{j!}{(j+2)!}} \frac{e^{\frac{n}{2\alpha}}}{\sqrt{\alpha^3}} \cdot K_j\left(\frac{n}{\alpha}\right), \quad (5)$$

where $j$ denotes the order of spherical Laguerre function, $\alpha > 0$ is a parameter which controls the rate of exponential decay of $d_j(n)$ and $K_j(n)$ is the generalized Laguerre polynomial of order 2 that is given by:

$$K_j(n) = \sum_{r=0}^{j} \binom{j+2}{j-r} \frac{(-n)^r}{r!}. \quad (6)$$

To obtain the basis set $\mathbf{B}(L_i,M)$ that was employed herein, the spherical Laguerre basis function $d_j(n)$ that is given by Eq. (5) was convolved with a Gaussian kernel $g(\tau,\sigma)$; $\tau > 0$, $\sigma = 1$

$$b_j(n) = d_j(n) * g(\tau, \sigma), \quad (7)$$

where $\tau > 0$ controls the pure time-delay of the hemodynamic response with respect to the neural event onset. To ensure orthogonality of $\mathbf{B}(L_i,M)$, the constructed basis functions $b_j(n)$ were orthogonalized with respect to each other using the Gram-Schmidt process.

The free parameters $L_i = 1,2,3,\dots$; $a > 0$; and $\tau > 0$, which uniquely determine a spherical Laguerre basis set $\mathbf{B}(Li,M)$, were determined for each subject separately yielding subject-specific basis sets. Model performance was evaluated in terms of the model generalization error that is based on the mean squared error, which is given by:

$$\text{mse}(\alpha, \tau) = \frac{1}{N} \sum_{n=1}^{N} (\hat{y}(n) - y(n))^2, \quad (8)$$

where $y(n)$ and $\hat{y}(n)$ denote respectively the measured and predicted BOLD signal time-series at times $n=1,\dots,N$. The model generalization error was computed as the average $\text{mse}(\alpha,\tau)$ obtained across all folds.

To prevent overfitting, the order $L_i$ of the BOLD signal model described by Eq. (3) was fixed to $L_i = L = 2$ for both experimental conditions ($i = 1,2$). The selection of the model order was determined based on the Bayesian information criterion. Optimal values for the $\alpha > 0$ and $\tau > 0$ parameters were determined based on the minimum generalization error using a grid search. In voxel-wise analyses, performing grid search to determine optimal values for these parameters incurs heavy computational burden. To reduce this, the voxel-wise analysis was performed in two steps: in the first step, an optimal basis set $\mathbf{B}(L,M)$ was constructed for each subject based on an ROI analysis using a preselected set of ROIs (see MRI data acquisition and pre-processing). The optimal parameter values for $\alpha > 0$ and $\tau > 0$ were determined based on the average generalization error obtained across all ROIs. To identify differences in the HRF curve shape between different ROIs and experimental conditions, we applied principal component analysis to the group of HRF estimates obtained across all participants, for each ROI and experimental condition. The HRF curve shape of each group was evaluated in terms of the first principal component, which accounted for most of the variance across all subject-specific HRF estimates contained within each group. In the second step, the optimal basis set $\mathbf{B}(L,M)$ that was constructed for each subject in the previous step was employed to obtain optimal voxel-specific HRF estimates for each experimental condition using Eq. (2).

The optimal voxel-specific HRF estimates obtained for each subject were used in a subsequent analysis to study neuronal activity in response to novelty versus repetition. To this end, a GLM was constructed for each subject using three regressors: one constant term for modeling the intercept, and two regressors each of which associated with a different experimental condition. The time-course of each condition was initially convolved with the corresponding condition-dependent HRF estimate. Subsequently, the derived regressors were z-transformed and entered into a GLM analysis. A parameter estimate associated with each condition was obtained using ordinary least squares regression (OLSR). It should

be noted that this approach provides more accurate parameter estimates, which could not be extracted directly from the first part of the analysis (modeling the BOLD signal and HRF estimation) nor from the estimated HRF shape. The advantage of this approach for quantifying the strength of the hemodynamic response to novelty or repetition events is that it takes the entire HRF into consideration as illustrated in Supplementary results 2.

**Generalized psychophysiological interaction (gPPI) fMRI analyses.** gPPI fMRI analysis was performed to study novelty-dependent FC of the LC with voxels within our preselected ROIs (see section "fMRI data analysis")[27]. To this end, a GLM was constructed for each subject using a total of 5 regressors: one physiological, two psychological, and two interaction regressors[88]. The physiological regressor was constructed as the average of all voxels within the LC ROI (seed region). The analysis was also repeated using an eroded version (60% volume reduction) of the LC ROI (Supplementary Fig. 8). The psychological regressors were constructed by convolving each of the task-dependent block time-series with the group-level LC HRF, which is shown in Fig. 2a and derived as described in section fMRI data analysis. This was important in order to align the block time-series associated with each condition with the physiological regressor in time. The interaction regressors were calculated by multiplying each of the psychological regressors with the physiological regressor. A parameter estimate associated with each regressor was estimated using OLSR.

**Statistical analyses.** Statistical analyses were performed in R (version 4.0.1; https://www.r-project.org/). Group characteristics were summarized in medians and interquartile ranges (IQRs). Differences between Aβ+ and Aβ− individuals were assessed using Kruskal-Wallis analysis of variance tests for continuous variables, and Chi-squared ($\chi^2$) tests for categorical variables, respectively.

Statistical parametric maps identifying voxels within the predefined set of ROIs with significant task-related BOLD signal or task-related FC values between the LC and other brain voxels (LC-FC) were generated using a two-level analysis: at the first level (subject level), conditions were contrasted against each other to create contrast images of Novel Face-Name Pairs versus Repeated Face-Name Pairs (NvR). At the second level (group level), statistical parametric maps were generated using the subject-specific NvR contrast images using LME modeling. Parameter estimation was performed using maximum likelihood estimation, containing fixed effects for the predictors of interest, random intercepts for participants, and random slopes for functional runs. In all LME models baseline age and sex were included as covariates. Models examined NvR-associated brain activity patterns and the effect modification by PiB. The analyses were performed using both continuous and dichotomous PiB.

To examine which voxels within the predefined set of ROIs exhibited NvR activity or FC with the LC in association with longitudinal PACC5 change, and potential effect modification by PiB, we used LMEs with the maximum likelihood estimation, containing fixed effects for the predictors of interest, random intercept for each subject, and random slope for time (number of years between baseline and follow-up cognitive assessments). In all LME models, baseline age, sex and years of education were included as covariates. We investigated the following hierarchical set of models: (i) PACC5 ~ NvR contrast × Time + covariates, (ii) PACC5 ~ NvR contrast × PiB × Time + covariates. Interactions of the covariates with time were excluded if $p > 0.10$ and removal of the interaction did not change the effect of other predictors. We provide the formula for the most complex model (with three-way interaction):

$$
\begin{aligned}
\text{Outcome}_{ij} = {} & \beta_0 + \beta_1 \text{Age}_i + \beta_2 \text{Sex}_i + \beta_3 \text{Edu}_i + \beta_4 \text{PredA}_i + \beta_5 \text{PredB}_i \\
& + \beta_6 \left( \text{Age}_i \times \text{Time}_{ij} \right) + \beta_7 \left( \text{Sex}_i \times \text{Time}_{ij} \right) + \beta_8 \big( \text{Edu}_i \\
& \times \text{Time}_{ij} \big) + \beta_9 \left( \text{PredA}_i \times \text{Time}_{ij} \right) + \beta_{10} \left( \text{PredB}_i \times \text{Time}_{ij} \right) \\
& + \beta_{11} \left( \text{PredA}_i \times \text{PredB}_i \times \text{Time}_{ij} \right) + b_{si} \text{Time}_{ij} + b_{0i} + e_{i,j}
\end{aligned}
\tag{9}
$$

where $i$ denotes participants, $j$ denotes measurements, $\text{Var}(b_{0i}) = \tau_0^2$, and $\text{Var}(b_{si}) = \tau_s^2$, $\text{Cov}(b_o b_s) = \rho \times \tau_0 \times \tau_s$. Outcome$_{ij}$ is the outcome variable measured over time (i.e., PACC5); Age$_i$, Sex$_i$ and Edu$_i$ indicate age, sex and years of education at baseline; Time$_{ij}$ (years) is the time at the follow-up testing session relative to the baseline session; $b_{0i}$ is the random intercept for each subject; $b_{si}$ is the random slope for each subject; $\tau_0^2$ is the variance of the residuals of the random intercept; $\tau_s^2$ is the variance of the residuals of the random slope; and $e_{i,j} \sim \mathcal{N}(0, \sigma \cdot \mathbf{I}_{i,j})$. $\mathbf{I}_{i,j}$ denotes the identity matrix and $\sigma$ the noise standard deviation. Pred A/B denote the predictor variables of interest (A = NvR contrast of either LC activity or LC-FC; B = PiB) depending on the investigated model. The analyses were repeated for dichotomous PiB.

The generated statistical parametric maps were adjusted for multiple comparisons using cluster-extent-based thresholding. For detecting greater brain activation or LC-FC during NvR we used a cluster defining threshold of $Z > 4.5$ with a two-tailed, FWER-corrected $p < 0.05$[89]. For identifying brain regions exhibiting significant associations between NvR activity (or LC-FC) and longitudinal PACC5 change we applied a cluster defining threshold of $Z > 3.1$ and two-tailed, FWER-corrected $p < 0.05$ to account for the different sources of variability associated with each data modality. For the statistical parametric maps derived from sensitivity analyses performed with smaller sample sizes, we used a

cluster defining threshold of $Z > 2.3$ and two-tailed, FWER-corrected $p < 0.05$. To provide a comprehensive view on potential subthreshold patterns of brain activation, we reported null-findings post-hoc using FDR-adjustment with a $q$-value of 0.05[90] in the Supplementary material.

To visualize the relationship between (i) LC activation, or (ii) LC-FC and PACC5 decline, we extracted the parameter estimates from the clusters of the cluster-extent thresholded maps. We performed floodlight analyses to determine the range of PiB values where NvR activation or LC-MTL FC were significantly associated with PACC5 decline. To understand whether the effects may be driven by certain cognitive domains, we repeated the longitudinal analyses for each of the subtest scores of the PACC5 battery, as well as the executive functioning and memory composite scores (adjusted for multiple testing using FDR with a $q$-value of 0.05[90]). Residual plots and QQ plots were examined for all models. All reported beta coefficients were unstandardized except in Figs. 5d and 7d, and $P$ values were two-sided.

**Reporting summary.** Further information on research design is available in the Nature Research Reporting Summary linked to this article.

## Data availability

The Harvard Aging Brain Study project is committed to publicly releasing its data. Baseline structural MRI, PiB-PET and cognitive follow-up data until year 5 is publicly available to the research community at http://nmr.mgh.harvard.edu/lab/harvardagingbrain/data. Task-fMRI data are currently not yet publicly available but will be made available in future releases. Requests for material, currently available raw and processed data for all the datasets used in the study, and correspondence can be addressed to Dr. Sperling. Qualified investigators must abide by the Harvard Aging Brain Study online data use agreement, designed to protect the privacy of our participants. Source data are provided with this paper.

## Code availability

All analyses were performed using the available toolboxes: R version 4.0.1 (http://www.r-project.org/), MATLAB R2018b (https://www.mathworks.com), FSL version 5.0.7 (https://fsl.fmrib.ox.ac.uk/fsl/fslwiki/), SPM12 https://www.fil.ion.ucl.ac.uk/spm/software/spm12/, FreeSurfer version 6 (http://freesurfer.net) and ANTs version 2.1.0 (http://stnava.github.io/ANTs/).

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

## Acknowledgements

We would like to thank all the participants of the Harvard Aging Brain Study. This work was supported in part by the Gordon Center for Medical Imaging P41 EB022544 and the Athinoula A. Martinos Center for Biomedical Imaging P41 EEB015896, as well as shared instrumentation grants: S10OD018035, S10RR021110, S10OD010364, S10RR023401, S10RR023043, and 1S10RR019307. This research was supported by the Harvard Aging Brain Study P01 AG036694 (MPIs Reisa Sperling, MD and Keith Johnson, MD), NIH grant R01 AG046396 (PI Keith Johnson, MD), NIH grant T32 EB013180 (PI El Fakhri Georges, PhD), NIH grant R01AG062559 and R01AG068062 (PI Heidi Jacobs, PhD) and Dekker-Padget Dutch2USA Grant (Nina Engels-Domínguez).

## Author contributions

P.C.P. and N.E.D. aided in study design, performed methods development, analyzed imaging data, performed statistical analyses, interpreted data, and wrote the manuscript. K.V.P. and D.M.R. aided in data collection, neuropsychological assessments, interpreted the results and reviewed the manuscript. N.E.D. and M.R.S. aided in data collection, study organization and data management. C.S. aided in methods development, interpreted the data and reviewed the manuscript. M.R.S., A.P.S., R.F.B. aided in MRI methods, data management and reviewed the manuscript. M.E.F and G.E.F. aided in PET methods, interpreted the results, and revised the manuscript. Y.T.Q. interpreted the results and revised the manuscript. R.A.S. and K.A.J. provided the participants, data analytic tools, aided in study design, interpreted results, and revised the manuscript. H.I.L.J., D.M.R., G.E.F, R.A.S., and K.A.J. acquired the financial support for the project leading to this publication. H.I.L.J. conceptualized this study, aided in study design, aided in methods development, aided in statistical analyses, interpreted data, revised the manuscript, and had the general supervision of the study.

## Competing interests

K.V.P. is funded by NIA grant K23 AG053422-01 and the Alzheimer's Association and has served as a paid consultant for Biogen. A.P.S. has been a paid consultant for Janssen, Biogen, Qynapse, and NervGen. D.M.R. has done consulting for Biogen, Idec and Digital Cognition Technologies and served on the Scientific Advisory Board for Neurotrack. R.F.B is funded by grants from the NIH K99/R00 (R00AG061238) and the Alzheimer's Association. Y.T.Q is funded by grants from the NIH NIA (R01 AG054671, R01AG066823), the Alzheimer's Association, and Massachusetts General Hospital ECOR, and has served as a paid consultant for Biogen. K.A.J. has served as paid consultant for Janssen, Genzyme, Novartis, Biogen, Roche, and AC Immune. He is a site co-investigator for Lilly/Avid and Janssen, and receives research support for clinical trials from Eisai, Lilly and Cerveau. K.A.J. received funding from NIH grants R01 EB014894, R21 AG038994, R01 AG026484, R01 AG034556, P50 AG00513421, U19 AG10483, P01 AG036694, R13 AG042201174210, R01 AG027435, and R01 AG037497 and the Alzheimer's Association grant ZEN-10-174210. RAS has served as a paid consultant for AC Immune, Acumen, Alnylam, Biogen, Cytox, Genentech, Ionis, Janssen, JOMDD, Neuraly, Neurocentria, Oligomerix, Prothena, Renew, Roche, Shionogi and receives research support for clinical trials from Alzheimer's Association, Eisai, Eli Lilly and Co. and NIA. She also receives research support from the following grant: P01 AG036694, U01 AG032438, U01 AG024904, R01 AG037497, R01 AG034556, K24 AG0350007, P50 AG005134, U19 AG010483, R01 AG027435, Fidelity Biosciences, Harvard Neuro-Discovery Center and the Alzheimer's Association. These relationships are not related to the content in the manuscript. All other authors report no relevant conflicts.
