## [Peer Review File · Nature Communications]

Reviewers' Comments:

Reviewer #1:

Remarks to the Author:

Prokopiou et al present a very interesting and thought-provoking manuscript entitled: "Lower novelty-related locus coeruleus function is associated with A β - related cognitive decline in clinically normal individuals". The authors present associations between fMRI BOLD signal in the locus coeruleus (LC) and in temporal brain regions in relation to novelty versus repetitive stimuli, as well as functional connectivity data between LC and the temporal ROIs. These parameters are further associated to PACC5 and a number of related cognitive measures and to PiB-derived A β -status (continuous and dichotomous). They conclude that lower novelty-related LC activity and functional connectivity between the LC and hippocampus and parahippocampal gyrus were associated with steeper decline in cognition, particularly in the presence of elevated A β deposition.

The manuscript is well written and main results are easy to follow. The results are novel and the theory of LC functional connectivity being affected early on in AD pathogenesis is interesting.

However, I have two major concerns related to the results presented:

1) The main area of interest in the brainstem (LC) is a very tiny nucleus, the size of a slightly elongated grain of rice (approximately 1x1.5x12 mm). The voxel size of the BOLD-MRI sequence is stated to be 3.1x3.1x5.0 mm. Thus the real volume of interest only constitutes at maximum 20% of the voxel volume where it is located. Further the total area of interest is contained in approximately two BOLD voxels. I understand that the authors have applied several measures to overcome these problems and to some extent the localization problem can be overcome by colocalizing the region using a more high-resolution sequence. But also using a standard 1x1x1 mm 3T MRI it is difficult to delineate the locus coeruleus. Getting a reliable signal from the locus coeruleus must be considered borderline of what can be accomplished using 3T fMRI.

2) Probably related to the small region of interest the results are, albeit interesting, not very consistent. In the novelty-related LC activity in relation to A β and PACC the right LC is indicated to be affected (Fig 5). In the subsequent FC experiment the left hippocampus and is involved, and bilateral hippocampi and left parahippocampal gyrus are indicated when A β is taken into consideration. Some results are FWER-corrected, some are FDR-corrected. There are also a large number of cognitive test comparisons made, and apart from the voxel-wise analyses there seems to be no correction for multiple comparisons. It may also be that the results are a bit inconsistent due to the limited number of participants on an AD trajectory. Only 36 participants are PiB positive out of a total of 128 participants, leaving 92 PiB negative.

For publication the results need to be replicated in an independent cohort to make sure that the findings are not just random false-positive results due to multiple comparisons and small group sizes.

Reviewer #2:

Remarks to the Author:

The manuscript by Prokopiou, Engels-Domínguez, et al. examines novelty-related activity and function connectivity with the locus coeruleus (LC), and how this relates to amyloid pathology and cognitive performance. The authors first find increased activation to novel stimuli in fronto-temporal cortex and LC. Furthermore, functional connectivity between the LC and medial temporal lobes is increased during presentation of novel stimuli. In the next set of analyses, they find that higher LC activity and greater functional connectivity is associated with better cognitive performance cross-sectionally, and better maintenance of performance over time. Interestingly, this effect was stronger in individuals with more amyloid.

This was a well-written manuscript with well-considered analyses and very interesting results. There has been increased focus on the role of the LC in early Alzheimer's progression, but studies have primarily focused on indices of structural integrity due to the difficulty in collecting fMRI in this region. Using methods particularly well-suited to investigating brainstem activity, they find LC function in the context of a novelty task that would be predicted based on prior literature. Importantly, the authors link the integrity of LC function to subsequent cognitive

performance in the presence of preclinical Alzheimer's pathology. The fact that preserved LC system functioning was related to maintained cognitive performance in the face of amyloid provides strong support for theories that the LC contributes to resilience, and its dysfunction may leave people vulnerable to pathology-related decline. I have a few suggestions that I believe will strengthen the manuscript, and a request for a few post-hoc test statistics to be removed (this would not affect conclusions, as the primary voxel-wise analyses provide appropriately controlled evidence). Otherwise, I think this is an extremely strong study that will be of high interest to many readers.

- Can the authors clarify if a separate t-test between conditions was run on values extract from significant clusters in Figures 3 and 4? If so, this statistical test should be removed as it is not independent from the method to select voxels. That is, a test between conditions in voxels selected because they showed significantly different activity between conditions would not be valid. The values in these voxels were already found to be significant at the whole brain level (and after adjusting for multiple correction), so presenting a descriptive plot of parameter estimates without an additional statistical test is sufficient. The same holds true for statistical tests of values from the 3-way interaction plotted in Figure 5B (main text lines 274, 276, 380, and 382).
- In the analysis of activity/LC-FC between Novelty vs Repeated was there an interaction effect with PIB?
- I appreciated the additional visualizations in Figure S6. I initially wanted to see these in the main text, or to have LC activity on the x-axis with different lines for PIB. However, I now see why the authors chose the presentation they did, with the blue box displaying the level of PIB at which the effect in LC became apparent.
- It would be helpful to specify in the caption of Figure 2 that each colored line in panel C represents a different subject (as opposed to different ROIs, as in the other panels).
- In the Methods section, should it say that motion correction was performed with MCFLIRT and artifact detection and removal with ICA-AROMA?

Reviewer #3:

Remarks to the Author:

The manuscript entitled, "Lower novelty-related locus coeruleus function is associated with A β -related cognitive decline in clinically normal individuals" adds to the growing field of functional neuroimaging of the locus coeruleus (LC) in human participants. There are many studies currently using neuromelanin-sensitive techniques to assess LC integrity in the context of neurodegenerative diseases. Fewer studies are assessing the activity and functional connectivity of the LC, much less in a clinical population, and fewer still are relating these outcomes to disease biomarkers. The current study addresses all of these important issues using cutting edge techniques to probe outcomes that could lead to better screening and treatment of neurodegenerative disorders like Alzheimer's disease (AD). The authors utilized the Harvard Aging Brain Study cohort to investigate the relationship between A β pathology, LC activity and functional connectivity, and cognitive outcomes in clinically normal older adults. The authors first demonstrated that activity and connectivity of the LC with the hippocampus and amygdala, as measured through fMRI, are higher under conditions of novelty. These measures were then used as a factor in associating to A β positron emission tomography status and cognitive decline. Evidence showed that lower novelty-related LC activity and connectivity with the hippocampus/parahippocampus were associated with steeper A β -related cognitive decline, and identified several key neurocognitive domains. Importantly, there were no direct effects of A β levels on measures of novelty-related activity or functional connectivity. These novel findings could have use as a potential biomarker for those who are at higher risk of developing AD, in addition to adding to the growing literature implicating the LC in the prodromal phase of AD.

Overall this is an elegant study that was well designed, executed, and presented. There are a few changes that would enhance the overall impact of this manuscript:

1. The authors go into great detail about links between A β and network hyperexcitability, but much of this is speculative and not necessarily LC-specific. For example, the authors cite papers demonstrating increases in CSF norepinephrine metabolite levels in AD, but the LC has not been

established as the source of CSF norepinephrine and metabolites, making links between such a measure and LC activity controversial. Otherwise, there is scant preclinical evidence that points to effects of AD pathology on LC firing, and no data was presented showing A β in the LC itself. Indeed, the work presented suggests the opposite, whereby lower LC has a detrimental effect on the relationship between AD pathology and cognitive measures. Thus, it is unclear where A β -induced hyperexcitability fits into the narrative – it seems unnecessary at best and counterintuitive at worst.

2. On a similar note, the authors should provide more discussion about the nature of the relationship between LC activity/functional connectivity and A β . Is A β causing LC dysregulation or vice versa? What might render the LC resilient to declines in response to novelty?

3. The idea of LC integrity encoding cognitive reserve seems to fit well with this study, and should be highlighted.

4. The authors spend a lot of time explaining that the LC develops early tau pathology, and tau is a better predictor of cognitive decline than plaques, so tau PET would have made sense. The rationale for assessing the relationship between A β PET and the LC is lacking.

5. The authors should discuss the potential confound of significant age differences between the A β positive and negative groups.

6. Figure 3B is missing asterisks indicating statistical significance.

There are also other minor changes that we would recommend or like clarification on:

1. In the abstract, line 32: change “harboring tau” to “harboring pretangle hyperphosphorylated tau”

2. In the intro, line 70: it would be beneficial to incorporate some recent articles establishing the role of the noradrenergic system and its forebrain targets in novelty responses (e.g. Lustberg et al. *Psychopharmacology*, 2020)

3. For Figure 2, inclusion of statistics for panel D would be helpful, as some of the box and whisker plots look almost identical but are stated as being significantly different (ie. PHG, EC, INS, and LC)

Reviewer #4:

Remarks to the Author:

The manuscript reports the results of a study examining if activity (and functional connectivity) of the locus coeruleus (LC) is associated with cognitive decline in a cohort of healthy older subjects (at baseline). The study also included PIB-PET measured at baseline, and examined if the association would be modulated by amyloid load. The brain activity was in response to a face-name task (novel faces versus familiar faces) and it was focused on specific regions previously known to be involved in memory, visual perception, and novelty. The study found that LC activity and LC functional connectivity between LC and amygdala and hippocampus was higher during novel face-name pairs versus familiar face-name pairs. Lower novelty related LC activity was associated with increased decline of cognition (associated with beta-amyloid load).

It is a very well designed study, very good statistical models and clearly presented.

Comments

In regards the BOLD signal analysis and the approach to defining the HRF for each participant and using that obtained HRF for analysis – isn't there a danger of circular analysis? You are using each task to define the HRF and then based on the HRF following with a GLM analysis. I would have expected that once obtained the HRF for each participant (and task) that the area under the curve (AUC) would have been calculated as a measure of activation level. What was the advantage of

your approach?

Given the small size of the LC – could partial volume effects be a significant issue? The voxel size in the BOLD signal were large relative the size of the LC. It has been found that the volume of the LC is associated with extent of tau- and Abeta pathology (smaller volume associated with higher pathology) – see for example doi: 10.3389/fnana.2017.00080, doi: 10.1016/j.brainres.2017.12.027 , doi: 10.3233/JAD-142445. Thus a smaller LC may mean a lower activation level and thus be a confound with the computed associations.

The LC is the major source of noradrenaline (NA) in the brain and is widely diffused across the brain, and in particular, underpins arousal, alertness and attention – see for example 10.1523/JNEUROSCI.1164-19.2019, doi: 10.1155/2017/6031478, doi: 10.1038/nrn2573. A theory of cognitive reserve gives prominent role to the right hemisphere (doi: 10.1016/j.neurobiolaging.2013.11.028) and it is quite interesting that there seems that stronger associations were found with this hemisphere and with tasks that are known to be right hemisphere dominant. It may be that novelty contributes to increased arousal and attention relative the familiar face-name pairs.

The results from figure 5(b) would seem to indicate that the decrease in PACC5 scores is quite slow (it seems about 0.5 SD in 7 years in most affected group) – would this level of decline be associated with conversion to MCI/AD. It is not clear from study if participants remained healthy status over the time frame of the study (within the time frame of each participant's follow up period). I am asking as the discussion makes reference to prodromal AD, and AD patients, yet as far as is reported, the participants are 'healthy' status over the time frame of the follow up period.

I am surprised that study does not include CSF based measures of tau and amyloid -that would have one way to include measures of the relevant pathology over time.

Comparing figures 5(b) and 7(b), the left hand side xy plot, I would have expected that the y-axis would show the same range in PACC5 values – one figure it is about 0.5 SD, and the other 1SD. There is a consistency in the right hand side plot between both figures.

The authors used global measures of ABeta, would perhaps measures of ABeta in the specific ROI would have been a better measure of ABeta load? Particularly since the objective of the study was to demonstrate an association between Abeta and LC activity or functional connectivity to the MTL.

Response letter: NCOMMS-21-27589

We thank the reviewers for their positive and excellent remarks and in-depth comments that allow us to emphasize the impact of our manuscript. Below are our responses to the comments of the reviewers. We have added additional sensitivity analyses to validate our main findings presented in the revised manuscript, which included analysis of (i) unsmoothed data, (ii) a different confirmatory dataset (*Replication Dataset*), (iii) a subset of our main cohort matched for age, sex and years of education (*Matched Dataset*), as well as (iv) controlling for grey matter density. We have also reworked our introduction to clarify our hypotheses and rationale for investigating the relationship between LC function and A β . In addition, we have added a more elaborate discussion on the unilaterality of our findings in relation to the concept of cognitive reserve. Changes to address each of the reviewers' concerns are colored in red in the manuscript. Furthermore, we provide our responses in a point-by-point style in blue and have copy-pasted our changes in the manuscript along with their location in the manuscript in the text boxes (in green) below each comment. Figures starting with *Fig. R#* are presented in the response document. Figures starting with *Fig. S#* are presented in the *Supplementary material*.

Reviewer #1 (Remarks to the Author):

Prokopiou et al present a very interesting and thought-provoking manuscript entitled: “Lower novelty-related locus coeruleus function is associated with A β - related cognitive decline in clinically normal individuals”. The authors present associations between fMRI BOLD signal in the locus coeruleus (LC) and in temporal brain regions in relation to novelty versus repetitive stimuli, as well as functional connectivity data between LC and the temporal ROIs. These parameters are further associated to PACC5 and a number of related cognitive measures and to PiB-derived A β -status (continuous and dichotomous). They conclude that lower novelty-related LC activity and functional connectivity between the LC and hippocampus and parahippocampal gyrus were associated with steeper decline in cognition, particularly in the presence of elevated A β deposition.

The manuscript is well written and main results are easy to follow. The results are novel, and the theory of LC functional connectivity being affected early on in AD pathogenesis is interesting.

However, I have two major concerns related to the results presented:

Comment 1: The main area of interest in the brainstem (LC) is a very tiny nucleus, the size of a slightly elongated grain of rice (approximately 1x1.5x12 mm). The voxel size of the BOLD-MRI sequence is stated to be 3.1x3.1x5.0 mm. Thus, the real volume of interest only constitutes at maximum 20% of the voxel volume where it is located. Further the total area of interest is contained in approximately two BOLD voxels. I understand that the authors have applied several measures to overcome these problems and to some extent the localization problem can be overcome by colocalizing the region using a more high-resolution sequence. But also using a standard 1x1x1 mm 3T MRI it is difficult to delineate the locus coeruleus. Getting a reliable signal

from the locus coeruleus must be considered borderline of what can be accomplished using 3T fMRI.

Reply: We thank the reviewer for acknowledging our efforts to improve localization. We indeed also discussed the native resolution of our fMRI data as one of the limitations of our study (Lines 537-558). However, instead of extracting timeseries from a locus coeruleus (LC) region-of-interest (ROI), we took a data-driven approach by examining novelty-related activity within the entire brainstem as well as other regions-of-interest. This was the case for both our main analyses: the novelty versus repetition (NvR) activity and the PACC-related analyses. These analyses revealed clusters in the brainstem, including the area of the LC. We then used a widely accepted probability template (validated with ex-vivo data ¹) to determine whether this activity pattern included the LC. Voxels that survived our threshold in the activity NvR analyses and were localized within this template were extracted for visualization of these associations using boxplots. Registration of this template was performed using ANTs diffeomorphic warping and visually checked in every subject. A symmetric group of contiguous voxels with the greatest probability of being in the LC across most subjects was used to define our LC ROI in the functional space. This procedure was performed to ensure we included those voxels with the highest likelihood of capturing the LC.

With regards to our LC-medial temporal lobe (MTL) functional connectivity (FC) analysis, we indeed extracted time courses from this probabilistic template. To mitigate any concerns regarding localization, we now replicated these results with an eroded version (Fig. R1 – Fig. S24 in the *Supplementary material*) to ensure an even greater likelihood of capturing the LC. These validation results obtained using the eroded LC ROI are shown in Fig. R3 (Fig. S6 in the *Supplementary data*). This result has been added in the *Results* section (Lines 241-243), *Methods* section (Lines 749-752), and also discussed in the *Discussion* section (Lines 545-550) of the revised manuscript.

In addition to our overall data-driven approach, we also implemented several preprocessing methods – as the reviewer also recognized - that bring our spatial resolution of the data to a comparable and sometimes even better resolution than other studies. Our ellipsoid smoothing approach fits with the shape of the locus coeruleus and also limits including cerebrospinal fluid (CSF)-signal into the LC. Where other studies had a resolution comparable or slightly better than ours, application of a spherical spatial smoothing in those studies resulted into a reduction of the original voxel size resolution to lower levels than in our study.

Here are examples of some excellent, recent studies examining LC function in different populations:

- Del Cerro et al., 2020, *Journal of Psychiatric Research* ²: voxel size = 3.75 x 3.75 x 4 mm – performed smoothing using a spherical Gaussian filter of 8 mm Full-Width Half-Maximum (FWHM) and seed-to-voxel analysis with a seed based on the Keren et al., (2009) LC atlas ¹.
- von der Gablentz et al., 2015, *Neuroscience* ³: voxel size = 3.125 x 3.125 x 1.5 mm – performed smoothing using a spherical Gaussian filter of 8 mm FWHM and localized activation within the LC based on the Keren et al., (2009) atlas ¹;
- Köhler et al., 2016, *Human Brain Mapping* ⁴: voxel size = 2.67 x 2.67 x 2.7 mm – performed smoothing using spherical Gaussian filter of 6 or 4 mm FWHM and localized

LC activation based on a high-resolution atlas template of the human brainstem and cerebellum⁵ provided by the SUI toolbox¹;

- Grueschow et al., 2021 *Nature Communications*⁶: voxel size = 2.5 x 2.5 x 2.5 mm – performed smoothing using spherical Gaussian filter of 6 mm FWHM and ROI analysis based on the Keren et al., (2009) atlas LC ROI¹;
- Munn et al., 2021, *Nature Communications*⁷: voxel size = 3 x 3 x 4 mm – performed smoothing using a spherical Gaussian filter of 8 mm FWHM and ROI analysis based on the Keren et al., (2009) atlas LC ROI¹.
- Yebra et al., 2019 *Nature Communications*⁸: voxel size = 3 x 3 x 2.2 mm – performed smoothing using spherical Gaussian filter of 6 mm FWHM and localized activation in the LC based on the Keren et al., (2009) atlas¹;
- Lee et al., 2018 *Nature Human Behaviour*⁹: voxel size = 4 x 4 x 4 mm – performed smoothing using spherical Gaussian filter of 5 mm FWHM and visually localized activation within the LC based on the Keren et al., (2009) atlas¹.

To quantify this: the ellipsoid Gaussian kernel employed in our study was constructed using $\sigma = 1.125 \cong 13.24$ mm (FWHM) for the major diameter and $\sigma = 0.6 \cong 4.23$ mm (FWHM) for the minor². Using these approximations for the major and minor diameters, the volume of our kernel is approximately 124 mm³. The volume of a spherical Gaussian kernel that was employed in the other above-mentioned studies is approximately 113.1 mm³ for a Gaussian filter of 6 mm FWHM, and approximately 268.1 mm³ for a Gaussian filter of 8 mm FWHM, which is nearly 2 times larger than the resolution of our data after preprocessing³. This discussion has been added in the *Discussion* section of the revised manuscript (Lines 545-547), and *Supplementary material* (Section *Supplementary results S1*).

Furthermore, to illustrate that the application of spatial smoothing did not bias our results, we repeated the detection analysis of brain activation during novelty versus repetition using unsmoothed data. This analysis provides maximum spatial resolution (48 mm³) at the cost of lower signal-to-noise ratio (SNR). The results (Fig. R2, Fig. R4 – Fig. S1, Fig. S7 in the *Supplementary material*) were similar to the results obtained using the processing procedure presented in the manuscript, suggesting that detection of activated brain areas during novelty was not affected by the amount of spatial smoothing that was applied in our analysis. Thus, even though the original voxel size resolution of our data is relatively large, our sensitivity analyses using unsmoothed data confirmed the validity and robustness of our findings presented in the original manuscript and

¹ This work also presented results obtained using unsmoothed data.

² The relationship between the standard deviation of a Gaussian kernel and FWHM (in mm) is approximately given by $FWHM = (\sigma\sqrt{8\ln 2}) \cdot (\text{voxel size (in mm)})$.

³ Volume calculations: For the ellipsoid smoothing kernel with 13.24 mm (FWHM) for the major diameter and 4.23 mm (FWHM) for the minor diameter the volume is given by $\frac{4}{3} \cdot \pi \cdot r_1 \cdot r_2 \cdot r_3 = \frac{4}{3} \cdot \pi \cdot 6.62 \cdot 2.115 \cdot 2.115 \approx 124 \text{ mm}^3$, where r denoted the radius of each diameter. For the spherical kernel with 6 mm diameter the volume is given by $\frac{4}{3} \cdot \pi \cdot r^3 = \frac{4}{3} \cdot \pi \cdot 27 \approx 113 \text{ mm}^3$.

underscore that the novelty associated with these findings is strong. This new analysis has been added in the *Methods* section (Lines 749-752), and a relevant discussion has been also added in the *Discussion* section of the manuscript (Lines 545-550). Also, to illustrate that the BOLD signal in the 4th ventricle and the LC is uncorrelated, we calculated pairwise Pearson's correlations for each participant and each functional run (Fig. R5 – Fig. S20 in the Supplementary material), which confirmed that our original findings are not biased by the 4th ventricle activity. This has also been added in the *Discussion* section of the revised manuscript (Lines 535-537). It should be noted that the voxel-size resolution of our data is comparable to the resolution of the 3T data (36 mm³) used in a recent study on the arousal system-related modulations of neural dynamics (Munn et al., (2021), *Nature Communications*)⁷. The main findings in this work were replicated using an independent 7T dataset suggesting that even though the imaging resolution of the LC at 3T is relatively low, it still provides valuable information related to LC function.

In the Results section on page 11 (first paragraph – lines 241-243)

Our sensitivity analyses reproduced these findings using BOLD time-series extracted from an eroded version of the LC ROI (Fig. S6, Fig. S24), unsmoothed data (Fig. S7), as well as the Replication Dataset (Fig. S8).

In the Discussion section on page 23 (second paragraph – lines 535-537)

However, pairwise Pearson's correlation between BOLD-fMRI time-series extracted from the LC and 4th ventricle ROIs confirmed that our findings are not biased by partial volume effects (Fig. S20).

In the Discussion section on page 24 (first paragraph – lines 545-550)

This ellipsoid smoothing brought the resolution of our data to a comparable and sometimes even better resolution than other LC studies using spherical smoothing (Supplementary results S1). Furthermore, our sensitivity analyses performed using unsmoothed data (providing a maximum spatial resolution), an eroded version of the LC ROI, and the Replication and Matched Datasets demonstrated the robustness and reproducibility of our findings.

In the Methods section on page 30 (first paragraph – lines 749-752)

To demonstrate spatial smoothing did not bias our results by introducing signal from the CSF, we performed sensitivity analyses for NvR brain activation or LC-MTL FC during NvR using unsmoothed data.

In the Methods section on page 34 (second paragraph – lines 875-877)

The analysis was also repeated using an eroded version (60% volume reduction) of the LC ROI (Fig. S24).

Supplementary results S1

The ellipsoid Gaussian kernel employed in our study was constructed using $\sigma = 1.125$ for the major diameter and $\sigma = 0.6$. The relationship between the standard deviation of a Gaussian kernel and FWHM (in mm) is approximately given by

$$FWHM = (\sigma\sqrt{8 \ln 2}) * (\text{voxel size (in mm)}).$$

Using this approximation, the major diameter is 13.24 mm (FWHM), and the minor is 4.23 mm (FWHM).

The volume of an ellipsoid with these dimensions for the major and minor diameters is given by

$$\frac{4}{3} \cdot \pi \cdot r_1 \cdot r_2 \cdot r_3 = \frac{4}{3} \cdot \pi \cdot 6.62 \cdot 2.115 \cdot 2.115 \approx 124 \text{ mm}^3,$$

where r denotes the radius of each diameter.

The volume of a spherical Gaussian filter with a diameter of 6 mm FWHM, which is frequently used in the literature^{2,3,6,7,10} is given by

$$\frac{4}{3} \cdot \pi \cdot r^3 = \frac{4}{3} \cdot \pi \cdot 3^3 \approx 113 \text{ mm}^3,$$

where r denotes the radius diameter. For a spherical Gaussian filter with a diameter of 8 mm FWHM the volume is approximately equal to 268.1 mm³, which is nearly 2 times larger than the resolution of our data after preprocessing.

- Original bilateral LC ROI (10 voxels)
- Eroded bilateral LC ROI (4 voxels)

Figure R1: The bilateral LC ROI used in the initially submitted manuscript (red) and the eroded version of the LC ROI used to replicate the findings (blue). The eroded LC ROI is reduced in volume by 60% compared to the original ROI.

Figure R2: Top panel: validation results for brain activation during novelty versus repetition obtained using unsmoothed data. Bottom panel: Fig. 3 in the submitted manuscript illustrating the same result obtained using data smoothing.

Figure R3 Top panel: validation results for novelty-related LC-MTL functional connectivity obtained using psychophysiological interaction analysis. The physiological regressor was defined using an eroded version of the LC ROI (Fig. R1 (Fig. S24 in the *Supplementary material*) – blue ROI). Bottom panel: Fig. 4 in the submitted manuscript illustrating the same result obtained using the original version of the LC ROI (Fig. R1 – red ROI).

Figure R2: Top panel: validation results for novelty-related LC-MTL functional connectivity obtained using unsmoothed data. Bottom panel: Fig. 4 in the submitted manuscript illustrating the same result obtained using data smoothing.

Figure R3: Histogram of pairwise Pearson's correlation coefficients (top) and p-values (bottom) obtained between LC and 4th ventricle ROI time-series (number of time-points $N = 127$) from all participants and functional runs (number of tests $n = 768$). No significant correlations were observed ($r(125) < 0.001$, $p > 0.9$ uncorrected).

Comment 2: Probably related to the small region of interest the results are, albeit interesting, not very consistent. In the novelty-related LC activity in relation to $A\beta$ and PACC the right LC is indicated to be affected (Fig 5). In the subsequent FC experiment the left hippocampus is involved, and bilateral hippocampi and left parahippocampal gyrus are indicated when $A\beta$ is taken into consideration. Some results are FWER-corrected, some are FDR-corrected. There are also a large number of cognitive test comparisons made, and apart from the voxel-wise analyses there seems to be no correction for multiple comparisons. It may also be that the results are a bit inconsistent due to the limited number of participants on an AD trajectory. Only 36 participants are PiB positive out of a total of 128 participants, leaving 92 PiB negative. For publication the results need to be replicated in an independent cohort to make sure that the findings are not just random false-positive results due to multiple comparisons and small group sizes.

Reply: We apologize if the corrections we performed were not clear. A clarification regarding the correction for multiple comparisons has been added in the *Methods* section (Lines 925-936). We also would like to clarify that all the results presented in the manuscript were family-wise error-rate (FWER)-corrected. When no clusters survived our FWER-adjustment, we provided the less stringent false discovery rate (FDR) correction in the supplemental data. Given that so few studies have investigated LC function, we feel it is important to be transparent and provide readers with a comprehensive view on potential subthreshold patterns of brain activation. This will enable the comparison of our findings to those of future studies and will facilitate the interpretation of

findings. For example, our unilateral (right-sided) findings for the associations between A β -related PACC decline and novelty-related LC activity do not change under the less strict FDR-correction (Fig. R6 – Fig. S13 in the *Supplementary material*). Thus, there is no subthreshold left-sided activation, which is important when interpreting the cognitive processes involved as well as the role of cognitive reserve (as also suggested by reviewer 3, comments 2 and 3, and reviewer 4, comment 3). Exploration of our unilateral finding using FDR has now been added in the Results section (Lines 299-300). Furthermore, all analyses of the cognitive subtests of the PACC were corrected for multiple comparisons, as acknowledged in the captions of Tables S4-S7. This has been clarified in the *Methods* section (Lines 941-944).

In the Results section on page 13 (fourth paragraph – lines 299-300)

The lateralization of these findings did not change under the less strict FDR-correction (Fig. S13).

In the Methods section on page 37 (first paragraph – lines 925-936)

*The generated statistical parametric maps were adjusted for multiple comparisons using cluster-extent based thresholding. For detecting greater brain activation or LC-FC during NvR we used a cluster defining threshold of $Z > 4.5$ with a two-tailed, FWER-corrected $p < 0.05$ ^{11, 12}. For identifying brain regions exhibiting significant associations between NvR activity (or LC-FC) and longitudinal PACC5 change we applied a cluster defining threshold of $Z > 3.1$ and two-tailed, FWER-corrected $p < 0.05$ to account for the different sources of variability associated with each data modality. For the statistical parametric maps derived from sensitivity analyses performed with smaller sample sizes, we used a cluster defining threshold of $Z > 2.3$ and two-tailed, FWER-corrected $p < 0.05$. To provide a comprehensive view on potential subthreshold patterns of brain activation, we reported null-findings post-hoc using FDR-adjustment with a q -value of 0.05¹³ in the *Supplementary material*.*

In the Methods section on page 38 (first paragraph – lines 941-944)

To understand whether the effects may be driven by certain cognitive domains, we repeated the longitudinal analyses for each of the subtest scores of the PACC5 battery, as well as the executive functioning and memory composite scores (adjusted for multiple testing using FDR with a q -value of 0.05¹³).

Figure R6: Top panel: validation results for voxel-wise analyses relating NvR activity, PiB and longitudinal PACC5 measurements. The results were corrected for multiple comparisons using FDR. Bottom panel: Fig. 5 in the manuscript showing the same results obtained using FWER correction.

The reviewer also mentioned that our $A\beta^+$ group is relatively small. The prevalence of $A\beta^+$ among cognitively normal older individuals is estimated between 13 and 30% for this age range¹⁴, consistent with the 28% in our study. This has been clarified in the *Methods* section (Lines 691-695). To avoid relying on results of (smaller) subgroups, we also analyzed all $A\beta$ -related analyses with $A\beta$ continuously. Results were similar when using $A\beta$ dichotomously or continuously.

Finally, the reviewer suggested to replicate our findings using an independent cohort. This is of course challenging given the uniqueness of this dataset (baseline task-related fMRI and PiB-PET and longitudinal cognitive measurements over a period of 10 years). Nevertheless, to replicate our results we analyzed our *Replication Dataset*. Details about this dataset are provided in the *Methods*

section (Lines 583-594). The results of this sensitivity analysis are shown in Fig. R7 (Fig. S2 in the *Supplementary material*) and Fig. R8 (Fig. S8 in the *Supplementary material*), and have been added in the *Results* section of the revised manuscript (Line 201, and Line 243). They suggest that the main findings of our work presented in the submitted manuscript are reproducible and robust. A relevant discussion has also been added in the *Discussion* section (Lines 547-550).

In the Results section on page 9 (second paragraph – lines 200-203)

Our sensitivity analyses reproduced these observations when the unsmoothed data (Fig. S1) and the Replication Dataset (Fig. S2) were used, as well as when grey matter density was included into the model as a covariate (Fig. S3).

In the Results section on page 11 (first paragraph – lines 241-243)

Our sensitivity analyses reproduced these findings using BOLD time-series extracted from an eroded version of the LC ROI (Fig. S6, Fig. S24), unsmoothed data (Fig. S7), as well as the Replication Dataset (Fig. S8).

In the Discussion section on page 24 (first paragraph – lines 547-550)

Furthermore, our sensitivity analyses performed using unsmoothed data (providing a maximum spatial resolution), an eroded version of the LC ROI, and the Replication and Matched Datasets demonstrated the robustness and reproducibility of our findings.

In the Methods section on page 25 (first paragraph – lines 583-594)

To validate our imaging results, we analyzed two different fMRI datasets. The first one, the Replication Dataset, consisted of fMRI data acquired from forty-one older individuals using an alternative version of the face-name associative task. Twenty-four individuals overlapped with the main cohort but were scanned four years later using an alternative version of the face-name associative task. The other seventeen participants joined HABS later in the study and their baseline imaging and cognitive measurements were not within one year from each other and were therefore excluded from the main sample. The characteristics of the Replication Dataset are provided in Table S1. The other dataset, the Matched Dataset, consisted of a subset of 36 A β - individuals who were matched to the 36 A β + individuals based on the age, sex and years of education distributions using propensity-based matching. The characteristics of the Matched Dataset are provided in Table S2.

In the Methods section on page 29 (first paragraph – lines 691-695)

Based on this cut-off value, 92 participants in this study were classified as low A β (A β -), and 36 participants (28.13% of the entire cohort) with elevated A β (A β +) at baseline. This is consistent with the estimated prevalence of A β + individuals among cognitive normal individuals ranging from 13% to 30%¹⁴.

Figure R7: Top panel: validation results for voxel-wise analyses of NvR activity obtained using the *Replication Dataset*. The results were corrected for multiple comparisons using cluster extent thresholding (number of participants $n = 41$; cluster defining threshold $Z > 2.3$, two-tailed $p < 0.05$, FWER-corrected). Bottom panel: Fig. 3 in the submitted manuscript illustrating the same results obtained using the original cohort. The validation results show activation in similar brain regions as in Fig. 3. It should be noted that the size of the *Replication Dataset* is 68% smaller compared to the original dataset that was used in our study, which explains the overall decrease in effect size compared to the results shown in Fig. 3.

Figure R8: Top panel: validation results for LC-medial temporal lobe (MTL) functional connectivity obtained using the *Replication Dataset*. The results were corrected for multiple comparisons using cluster extent thresholding (number of participants $n = 41$; cluster defining threshold $Z > 2.3$, two-tailed $p < 0.05$, FWER-corrected). Bottom panel: Fig. 4 in the submitted manuscript illustrating the same result obtained using the original cohort. The validation results show significant functional connectivity of the LC with similar brain region as in Fig. 4. It should be noted that the size of the *Replication Dataset* is 68% smaller compared to the original dataset that was used in our study, which explains the overall decrease in effect size compared to the results shown in Fig. 4.

Reviewer #2 (Remarks to the Author):

The manuscript by Prokopiou, Engels-Domínguez, et al. examines novelty-related activity and function connectivity with the locus coeruleus (LC), and how this relates to amyloid

pathology and cognitive performance. The authors first find increased activation to novel stimuli in fronto-temporal cortex and LC. Furthermore, functional connectivity between the LC and medial temporal lobes is increased during presentation of novel stimuli. In the next set of analyses, they find that higher LC activity and greater functional connectivity is associated with better cognitive performance cross-sectionally, and better maintenance of performance over time. Interestingly, this effect was stronger in individuals with more amyloid.

This was a well-written manuscript with well-considered analyses and very interesting results. There has been increased focus on the role of the LC in early Alzheimer's progression, but studies have primarily focused on indices of structural integrity due to the difficulty in collecting fMRI in this region. Using methods particularly well-suited to investigating brainstem activity, they find LC function in the context of a novelty task that would be predicted based on prior literature. Importantly, the authors link the integrity of LC *function* to subsequent cognitive performance in the presence of preclinical Alzheimer's pathology. The fact that preserved LC system functioning was related to maintained cognitive performance in the face of amyloid provides strong support for theories that the LC contributes to resilience, and its dysfunction may leave people vulnerable to pathology-related decline. I have a few suggestions that I believe will strengthen the manuscript, and a request for a few post-hoc test statistics to be removed (this would not affect conclusions, as the primary voxel-wise analyses provide appropriately controlled evidence). Otherwise, I think this is an extremely strong study that will be of high interest to many readers.

Reply: We thank the reviewer for these positive comments and for appreciating our efforts to understand the role of LC function in AD pathophysiology, and providing suggestions that will help us improve the presentation of our findings and the clarity of the manuscript.

Comment 1: Can the authors clarify if a separate t-test between conditions was run on values extract from significant clusters in Figures 3 and 4? If so, this statistical test should be removed as it is not independent from the method to select voxels. That is, a test between conditions in voxels selected because they showed significantly different activity between conditions would not be valid. The values in these voxels were already found to be significant at the whole brain level (and after adjusting for multiple correction), so presenting a descriptive plot of parameter estimates without an additional statistical test is sufficient. The same holds true for statistical tests of values from the 3-way interaction plotted in Figure 5B (main text lines 274, 276, 380, and 382).

Reply: This is an excellent point. Our post-hoc statistical tests of the ROI parameter estimates for each condition shown in Fig. 3B and 4B in the manuscript are not independent from the method used to select voxels. Therefore, the t-test results have been removed from the revised manuscript.

Comment 2: In the analysis of activity/LC-FC between Novelty vs Repeated was there an interaction effect with PiB?

Reply: We thank the reviewer for his/her question. We apologize if this was not clear. For the analysis of activity/LC-MTL functional connectivity, adding PiB to the model as a covariate did not modify the activity or LC-MTL FC patterns between novelty and repetition. Furthermore, PiB

did not interact with NvR on activity/functional connectivity (Lines 207 - 208, and Lines 246 - 247).

In the Results section on page 9 (second paragraph – lines 207-208)

Adding PiB as covariate to the model did not modify the patterns of NvR activity. In addition, PiB did not interact with NvR on brain activation.

In the Results section on page 11 (first paragraph – lines 246-247)

Further, adding PiB as covariate to the model did not modify the patterns of NvR LC-FC and PiB did not interact with NvR on LC-FC.

Comment 3: I appreciated the additional visualizations in Figure S6. I initially wanted to see these in the main text, or to have LC activity on the x-axis with different lines for PIB. However, I now see why the authors chose the presentation they did, with the blue box displaying the level of PIB at which the effect in LC became apparent.

Reply: We thank the reviewer for this comment. Indeed, given the early involvement of the LC in the pathophysiology of Alzheimer's disease (AD), we also wanted to illustrate clearly to the readers at which PiB levels our associations between LC function and PACC decline act. We made sure in the initially submitted manuscript to also refer the readers to Figure S15 (Fig. S6 in the initially submitted manuscript).

In the Results section on page 14 (first paragraph – lines 306-307)

Fig. S15 illustrates the same result using dichotomous PiB.

Comment 4: It would be helpful to specify in the caption of Figure 2 that each colored line in panel C represents a different subject (as opposed to different ROIs, as in the other panels).

Reply: We thank the reviewer for this comment. We have added a legend in the the subplots in panel C (Fig. R9C) to make this more clear.

In the Fig. 2 caption

(C) Representative subject-specific HRF estimates obtained for the LC during Novelty (top panel) and Repetition (bottom panel) for 40 representative participants (each color represents a different, randomly chosen participant). The x-axis is the time [seconds]. The y-axis is the BOLD signal intensity [a.u.].

Figure R9: Updated version of Fig. 2 in the manuscript illustrating the regional variability of the hemodynamic response function with a legend added in (C) to indicate that each curve corresponds to a different participant.

Comment 5: In the Methods section, should it say that motion correction was performed with MCFLIRT and artifact detection and removal with ICA-AROMA?

Reply: We thank the reviewer for this comment. The following clarification has been added in the manuscript (Lines 725-729).

In the Methods section on page 30 (first paragraph – lines 725-729)

The basic pre-processing of the BOLD images was carried out using the Oxford Centre for Functional Magnetic Resonance Imaging of the Brain Software Library (FMRIB, UK – FSL version 5.0.7) ¹⁵. This included brain extraction, slice timing correction, motion correction via volume realignment, normalization to the 2 mm³ MNI-152 EPI template, and further artifact detection and removal using ICA-AROMA ¹⁶.

Reviewer #3 (Remarks to the Author):

The manuscript entitled, “Lower novelty-related locus coeruleus function is associated with A β -related cognitive decline in clinically normal individuals” adds to the growing field of functional neuroimaging of the locus coeruleus (LC) in human participants. There are many studies currently using neuromelanin-sensitive techniques to assess LC integrity in the context of neurodegenerative diseases. Fewer studies are assessing the activity and functional connectivity of the LC, much less in a clinical population, and fewer still are relating these outcomes to disease biomarkers. The current study addresses all of these important issues using cutting edge techniques to probe outcomes that could lead to better screening and treatment of neurodegenerative disorders like Alzheimer’s disease (AD). The authors utilized the Harvard Aging Brain Study cohort to investigate the relationship between A β pathology, LC activity and functional connectivity, and cognitive outcomes in clinically normal older adults. The authors first demonstrated that activity and connectivity of the LC with the hippocampus and amygdala, as measured through fMRI, are higher under conditions of novelty. These measures were then used as a factor in associating to A β positron emission tomography status and cognitive decline. Evidence showed that lower novelty-related LC activity and connectivity with the hippocampus/parahippocampus were associated with steeper A β -related cognitive decline and identified several key neurocognitive domains. Importantly, there were no direct effects of A β levels on measures of novelty-related activity or functional connectivity. These novel findings could have use as a potential biomarker for those who are at higher risk of developing AD, in addition to adding to the growing literature implicating the LC in the prodromal phase of AD.

Overall this is an elegant study that was well designed, executed, and presented. There are a few changes that would enhance the overall impact of this manuscript:

Reply: We thank the reviewer for these positive comments, for emphasizing the importance of investigating LC function in AD, and providing suggestions that will enhance the clarity and presentation of our manuscript.

Comment 1: The authors go into great detail about links between A β and network hyperexcitability, but much of this is speculative and not necessarily LC-specific. For example, the authors cite papers demonstrating increases in CSF norepinephrine metabolite levels in AD, but the LC has not been established as the source of CSF norepinephrine and metabolites, making links between such a measure and LC activity controversial. Otherwise, there is scant preclinical evidence that points to effects of AD pathology on LC firing, and no data was presented showing Ab in the LC itself. Indeed, the work presented suggests the opposite, whereby lower LC has a detrimental effect on the relationship between AD pathology and cognitive measures. Thus, it is unclear where A β -induced hyperexcitability fits into the narrative – it seems unnecessary at best and counterintuitive at worst.

Reply: We thank the reviewer for this valuable comment. We apologize if our narrative was confusing. The LC is the primary source of norepinephrine to the brain and in this context these CSF-studies examining norepinephrine (NE) and MHPG (metabolite of NE) can be relevant to our work and we aimed to provide a potential hypothesis that could clarify the different findings in CSF-studies and our own work. However, we do understand the comment of the reviewer: MHPG

may not necessarily reflect LC activity, but can also reflect function of the adrenergic receptors elsewhere in the brain. Therefore, to improve the narrative, we removed this section from our discussion and expanded our rationale and discussion clarifying the rationale for investigating the link between amyloid (A β) and LC function (Lines 49-69 and Lines 93-97).

In the Introduction section on page 3 (first, second and third paragraph – lines 49-69)

Several in-vitro and human autopsy studies suggested that one of the earliest sites in the brain implicated in these AD-related proteinopathies is the locus coeruleus (LC), a subcortical nucleus providing norepinephrine to the entire brain^{17, 18, 19}. While many of these studies focused predominantly on the role of tau, LC neurons and their axonal terminals can accumulate soluble, oligomeric variants of A β early on, which interact with tau deposits in the LC and which subsequently trigger aggregation of soluble and extracellular A β into plaques in the remote cortex during the initial phases of AD²⁰.

As both the LC and A β undergo early pathologic alterations, there may be a common mechanism contributing to the initial clinical symptoms of AD. These initial symptoms emerge typically after age 60 when cortical fibrillar A β is detectable, tau is omnipresent in the LC and has reached the hippocampus²¹. Recent animal work demonstrated that oligomeric A β in the LC has the capacity to dysregulate LC activity promoting early hyperexcitability²², which is reminiscent to the previously reported A β -driven excitatory toxicity in the cortical and hippocampal networks²³. As the disease progresses and tau accrues, neuronal hyperexcitability was followed by neuronal silencing in transgenic mice^{24, 25} or loss of hyperconnectivity in fMRI studies²⁶.

Thus, a closer investigation of the effect of A β on activity of the LC and its functional connectivity (FC), in particular with the medial temporal lobe (MTL) could improve our understanding of the evolution and the early detection of AD-related cognitive decline and provide new anchor points for interventions.

In the Introduction section on page 4 (second paragraph – lines 93-97)

Based on the staging of pathology, suggesting proliferation of cortical tau in these older individuals²¹, and its association with activity patterns, we hypothesized that novelty-related LC activity and connectivity would be lower at higher levels of A β , and that this lower novelty-related activity would be associated with A β -related cognitive decline.

Comment 2: On a similar note, the authors should provide more discussion about the nature of the relationship between LC activity/functional connectivity and A β . Is A β causing LC dysregulation or vice versa? What might render the LC resilient to declines in response to novelty?

Reply: We thank the reviewer for this question. The reviewer wonders about the nature of the directionality between A β and LC function. Our data is observational and as such does not allow to investigate causality. We should note that studies examining the earliest accumulation of A β

and formulating the pathophysiologic sequence are also limited by the detection abilities of the method or biomarker in use. Animal work has suggested that oligomeric A β may accumulate early and interact with the initial tau deposits in the LC. Muresan and Muresan²⁰ demonstrated that LC neurons and their axonal terminals can accumulate A β and may initiate spreading of oligomeric A β and aggregation into plaques in the remote cortex during the initial phases of AD. Recent animal work demonstrated that this oligomeric A β in the LC has the capacity to dysregulate LC activity, providing evidence for at least one direction: the impact of A β on LC neuronal activity²². The mechanism underlying the increased hyperexcitability of the LC may be related to the effect oligomers can have on alpha2-adrenergic receptors, leading to propagation of AD-pathology and cognitive decline in animals²⁷. This confirms that oligomers of A β may affect LC function, as also reported by Kelly et al. (2021), but provides evidence that their combined effect contributes to disease progression (which is consistent with our observations). See also comment 1 for how these findings fit within our hypotheses and findings.

The factors that may confer resiliency are still not clear. The LC has been suggested to be an important part of the biological substrate of the brain mediating cognitive reserve: a set of variables including education, intelligence, physical and mental activity, and environmental enrichment that allow the brain to adapt to AD-related pathology changes and maintain cognitive function^{28, 29, 30, 31, 32}. Environmental enrichment has been shown to improve cognitive function in mice³³, and cognitively stimulating activity has been associated with reduced A β (measured *in-vivo* using PiB-PET imaging) and a lower risk of dementia-related brain atrophy in older adults^{34, 35}. Animal research has suggested that greater novelty-related LC activity may be a potentially important component mediating the cognitive effects promoting cognitive reserve^{33, 36} via molecular mechanisms such as β -adrenergic enhanced neurogenesis²⁸ and elevated expression of plasticity-related genes³⁷. Therefore, environmental enrichment and cognitively stimulating activity might be potential factors that help to withstand pathology-related declines in the LC's response to novelty. Not all the factors contributing to cognitive reserve are collected objectively in HABS, which unfortunately does not allow us to examine the role of cognitive reserve in LC resiliency. This interesting discussion has been added in Lines 522 – 529 in the revised manuscript.

In the Discussion section on page 23 (first paragraph – lines 522-529)

Which LC-related factors may confer resiliency to AD pathology are not yet clear and warrant further examination. Animal research has suggested that greater novelty-related LC activity may be a potentially important component mediating the cognitive effects promoting cognitive reserve³³ via molecular mechanisms such as β -adrenergic enhanced neurogenesis²⁸ and elevated expression of plasticity-related genes^{33, 37} via molecular mechanisms such as β -adrenergic enhanced neurogenesis²⁸ and elevated expression of plasticity-related genes³⁷.

Comment 3: The idea of LC integrity encoding cognitive reserve seems to fit well with this study and should be highlighted.

Reply: We thank the reviewer for this question. Cognitive reserve is a concept explaining the individual variability in how brain pathology can affect cognitive abilities. In this framework, additional resiliency to pathology can come from education or exposure to cognitively stimulating

environments^{31, 32}. It has been hypothesized that the LC is an important part of the biological substrate in the brain mediating cognitive reserve²⁹. In support of this hypothesis, Clewett et al., (2009)³⁸ provided evidence that variations in LC structural integrity are associated with proxies of cognitive reserve, such as education, occupational complexity and verbal intelligence. Our analysis revealed that lower activation in the right LC evaluated using BOLD-fMRI is associated with steeper A β -related cognitive decline. Intriguingly, at higher levels of A β , we also noted individuals who were able to maintain higher LC function (either activation or FC) were also able to maintain their cognitive performance over time. Even though, these results were obtained after adjusting for years of education, this suggests that LC-related functional network organization can provide additional cognitive reserve even in the context of A β , consistent with its previously suggested neuroprotective role. The right lateralization observed in our findings is consistent with the involvement of the LC in cognitive processes associated with cognitive reserve, such as novelty, attention and arousal^{30, 39, 40, 41, 42, 43}. These processes have been shown to activate predominantly right fronto-parietal regions in the brain⁴⁴. Thus, maintenance of novelty-related activation of the LC might be indicative of fronto-parietal network reorganization promoting resiliency against the underlying pathology. This discussion has been added into the manuscript (Lines 467-482).

In the Discussion section on page 21 (second paragraph – lines 467-482)

It is known that beyond the LC's effects on learning⁴⁵, its putative function involves fast disengagement from specific tasks and amplification of attentional focus to the goal-relevant information. Additionally, allocation of attention has generally been ascribed to the right hemisphere⁴⁶, which aligns with the abundant involvement of attention in the digit symbol substitution test and our observation of lower right-sided LC activity being associated with steeper A β -related cognitive decline. This right lateralization is also consistent with the hypothesis that the LC may represent an important biological substrate underlying cognitive reserve and its associated processes, such as arousal, attention and novelty^{29, 43}, which all have been related to activation of predominantly the right fronto-parietal network^{30, 44}. Previous imaging work showed lower connectivity between the LC and frontoparietal or salience networks resulting in greater distractibility in older individuals^{47, 48}. Our results indeed demonstrated that individuals who are able to maintain optimal levels of novelty-related LC functional properties may be more resilient to cognitive decline, even in the presence of elevated A β . Together, these findings strengthen the role of the LC-NE system in modulating networks, promoting cognition and potentially supporting cognitive reserve.

Comment 4. The authors spend a lot of time explaining that the LC develops early tau pathology, and tau is a better predictor of cognitive decline than plaques, so tau PET would have made sense. The rationale for assessing the relationship between A β PET and the LC is lacking.

Reply: We thank the reviewer for this comment and apologize if the rationale for assessing the relationship between A β PET and the LC was not clear. Tau data acquisition was introduced in HABS during the 4th year of the study. Thus, we unfortunately did not have baseline tau data acquired within the same year as the baseline fMRI data, which would allow us to investigate the

relationship between LC function and tau directly. This is also acknowledged in the limitation section in the manuscript (Lines 530 - 533). We are currently collecting this data and hope to report on this in the future. We do understand that this may have confused the rationale and we have rewritten the introduction to clarify the importance of investigating the relationship between A β PET and the LC, as we also discussed in *Comment 1 and 2*.

In the Discussion section on page 23 (second paragraph – lines 530-533)

This study has limitations. First, as the LC is one of the first regions affected by tau, it would have been interesting to examine the relationship between FC and cortical tau deposition. Unfortunately, tau-PET imaging was recently introduced in HABS, adding analytical complexities in terms of varying time difference with the fMRI data.

Comment 5: The authors should discuss the potential confound of significant age differences between the A β positive and negative groups.

Reply: We thank the reviewer for this question. Indeed, there was an age-difference between the A β + and A β - groups. To account for effects of age, we included age as covariate in all our statistical models. No significant associations were observed between NvR brain activation and age after correcting for multiple comparisons using cluster-extent thresholding (number of participants $n = 128$; cluster defining threshold $Z > 4.5$, two-tailed $p < 0.05$, FWER-corrected). Similarly, no significant associations were observed between NvR LC-FC and age after correcting for multiple comparisons using cluster-extent thresholding. With regards to the relationship between PACC decline and NvR LC activity or LC-FC, we observed significant associations between age and the time courses of all ROIs of interest for the activity data, and in the bilateral amygdala and hippocampus for LC-FC analyses.

Given that age-adjustment is of course a statistical correction and does not remove all its possible contributions, we repeated the analysis for detecting significant NvR LC activity and LC-FC using the *Matched Dataset*: a subset of our A β - individuals who were matched to our 36 A β + based on the age, sex and years of education distributions using propensity-based matching.

With regards to NvR brain activity, the results (Fig. R10 – Fig. S4 in the *Supplementary material*) revealed significant brain activation in similar areas as the original results presented in the manuscript (Fig. 3). As expected, the effect size was smaller as the sample size is also smaller (~44% sample size reduction). The results obtained for LC-FC (Fig. R11 – Fig. S9 in the *Supplementary material*) were very similar to the results shown in Fig. 4 in the manuscript. As such, we can infer that age did not confound our findings.

Following the reviewer's suggestion, our results on the potential confounding effect of the age differences between the A β + and A β - have been added in the results section (Lines 204-207, and 244 – 247).

In the Results section on page 9 (second paragraph – lines 204-207)

Given the age difference observed between the $A\beta^+$ and $A\beta^-$ groups (Table 1) we post-hoc also repeated the same analysis using the Matched Dataset. The results (Fig. S4) revealed greater activation during NvR in similar areas as the results obtained for the entire cohort shown in Fig. 3A.

In the Results section on page 11 (first paragraph – lines 244-247)

Similar FC maps were obtained using the Matched Dataset (Fig. S9) compared to the results obtained using our original dataset shown in Fig. 4A. Further, adding PiB as covariate to the model did not modify the patterns of NvR LC-FC and PiB did not interact with NvR on LC-FC.

Figure R10: Top panel: validation results for brain activation during novelty versus repetition obtained using the Matched Dataset that includes 36 $A\beta^+$ and 36 $A\beta^-$ individuals with approximately equal distributions of age, sex and years of education. The selection of the $A\beta^-$

individuals was performed using propensity-based matching. Bottom panel: Fig. 3 in the submitted manuscript illustrating the same result obtained using the entire cohort.

Figure R11: Top panel: validation results for LC-FC during novelty versus repetition obtained using the *Matched Dataset* that includes 36 A β ⁺ and 36 A β ⁻ individuals with approximately equal distributions of age, sex and years of education. The selection of the A β ⁻ individuals was performed using propensity-based matching. Bottom panel: Fig. 4 in the submitted manuscript illustrating the same result obtained using the entire cohort.

Comment 6: Figure 3B is missing asterisks indicating statistical significance.

Reply: As suggested previously in *Comment 1 of Reviewer 2*, to not double dip in our results, we should not perform a formal statistical comparison of the parameter estimates obtained in each condition as the statistical significance was already evaluated in the voxel-wise analysis. Therefore, the asterisks in Fig. 3B and Fig. 4B have been removed from the revised manuscript.

There are also other minor changes that we would recommend or like clarification on:

Comment 7. In the abstract, line 32: change “harboring tau” to “harboring pretangle hyperphosphorylated tau”

Reply: We thank the reviewer for this clarification. Line 32 in the abstract has been changed in the revised manuscript as follows:

In the Abstract on page 2 (lines 31-33)

The locus coeruleus (LC), one of the initial subcortical regions harboring pretangle hyperphosphorylated tau, has widespread connections to the cortex modulating cognition.

Comment 8. In the intro, line 70: it would be beneficial to incorporate some recent articles establishing the role of the noradrenergic system and its forebrain targets in novelty responses (e.g. Lustberg et al. Psychopharmacology, 2020)

Reply: We thank the reviewer for providing these references. We have added these in the revised manuscript (Lines 71-74).

In the Introduction on page 3 (fourth paragraph - lines 71-74)

Animal studies have shown that novel or unexpected stimuli elicit phasic spikes in LC neurons, leading to NE release that targets the task-relevant regions in the brain, such as the amygdala and hippocampus in the MTL and the prefrontal cortex^{49, 50, 51}.

Comment 9. For Figure 2, inclusion of statistics for panel D would be helpful, as some of the box and whisker plots look almost identical but are stated as being significantly different (ie. PHG, EC, INS, and LC).

Reply: We thank the reviewer for this this suggestion. The asterisks have been added in Fig. 2D, which shows statistical comparisons of the HRF amplitude (HRF peak values) obtained within ROIs during NvR. Also, detailed statistics are provided in Table S3 in the *Supplementary material*. It should be noted that this suggestion refers to a different figure than *Comment 1 from Reviewer 2*, and *Comment 6 from Reviewer 3*, which refer to Figs. 3B and 4B.

Reviewer #4 (Remarks to the Author):

The manuscript reports the results of a study examining if activity (and functional connectivity) of the locus coeruleus (LC) is associated with cognitive decline in a cohort of healthy older subjects (at baseline). The study also included PIB-PET measured at baseline and examined if the association would be modulated by amyloid load. The brain activity was in response to a face-name task (novel faces versus familiar faces), and it was focused on specific regions previously known to be involved in memory, visual perception, and novelty. The study found that LC activity

and LC functional connectivity between LC and amygdala and hippocampus was higher during novel face-name pairs versus familiar face-name pairs. Lower novelty related LC activity was associated with increased decline of cognition (associated with beta-amyloid load).

It is a very well designed study, very good statistical models and clearly presented.

Reply: We thank the reviewer for these positive comments and constructive suggestions.

Comment 1: In regards the BOLD signal analysis and the approach to defining the HRF for each participant and using that obtained HRF for analysis – isn't there a danger of circular analysis? You are using each task to define the HRF and then based on the HRF following with a GLM analysis. I would have expected that once obtained the HRF for each participant (and task) that the area under the curve (AUC) would have been calculated as a measure of activation level. What was the advantage of your approach?

Reply: We thank the reviewer for this question. We respectfully disagree with the suggestion that our analysis is circular, because the second step (using the estimated hemodynamic response function (HRF) in a general linear model (GLM) analysis) was performed to obtain a parameter estimate quantifying the strength of the hemodynamic response to novelty and repetition stimuli. This estimate could not be computed directly during the first step of the analysis (HRF estimation) or extracted from the estimated HRF shape. Not including this second step would have biased our results, which we will explain further.

The parameter estimate (regression coefficient) obtained from a GLM analysis is a more suitable measure of the strength of the hemodynamic response to novelty or repetition events compared to features (e.g area under the curve (AUC)) extracted from the estimated HRF for each condition for the following reasons:

1. The parameter estimate obtained using GLM analysis uses information associated with the entire shape of the HRF rather than information isolated in individual time-points, such as the HRF peak and time-to-peak.
2. The physiological interpretation of features extracted from the HRF (peak value, time-to-peak, power, AUC) is not straightforward⁵². Moreover, some of these features are possibly related with properties of the vasculature rather than the underlying response to neuronal activation, such as vascular elastance and compliance^{53, 54}.
3. The GLM framework makes our analysis compatible with the vast majority of similar fMRI studies in the literature.

In relation to point 1, an HRF feature that might be more associated with the entire HRF curve is the AUC, as also mentioned by the reviewer. However, AUC corresponds to the steady-state step-

response to a step change⁴ in neuronal activation (Fig. R12A – Fig. S25A in the *Supplementary material*), which is different than the time-course of our task (mixed event-related and block design). In addition, it is often the case that HRF curves with different amplitude or shape characteristics have a similar AUC. For these reasons, we believe that AUC is not a suitable measure of the strength of the hemodynamic response to neuronal activation within the context of this study.

To illustrate this more clearly, consider a hypothetical experiment during which a subject is presented with two novelty events of which the timings are shown in Fig. R12B. A hypothetical BOLD response in a task relevant region (ROI 1 - Fig. R12C) and a task irrelevant region (ROI 2 – Fig. R12C) are shown in Fig. R12D, where the amplitude of the BOLD response in ROI 1 is four times greater than in ROI 2. The HRF estimates obtained using the BOLD signal responses in each ROI and the event timings are shown in Fig. R12E. In this case, the shape of the HRF curve is the same for both ROIs. The AUC is equal to 0 for both ROIs, even though the BOLD response in ROI 1 is four times greater than in ROI 2. In contrast to this, using the GLM framework as described in the manuscript gives a parameter estimate for ROI 1 that is four times greater than the parameter estimate for ROI 2.

This example illustrates how the GLM analysis approach is more advantageous for quantifying the strength of the hemodynamic response compared to the AUC extracted from the estimated HRF. We believe that understanding the regional variability of all the HRF features and their relationship with AD pathology is a very interesting research question that we aim to investigate in a future study including fMRI data collected with a higher temporal resolution.

This clarification has been added in Lines 863 - 868 in the manuscript and in subsection *Supplementary results S1* in the *Supplementary material*.

In the Methods section on page 35 (first paragraph – lines 863-868)

It should be noted that this approach provides more accurate parameter estimates, which could not be extracted directly from the first part of the analysis (modeling the BOLD signal and HRF estimation) nor from the estimated HRF shape. The advantage of this approach for quantifying the strength of the hemodynamic response to novelty or repetition events is that it takes the entire HRF into consideration as illustrated in Supplementary results S2.

Figure R12 (next page): (A) Illustration of the relationship between the AUC of the HRF and the steady-state step response of a linear hemodynamic model. The step response is obtained as the convolution of the hemodynamic response function with a step function and consist of a transient

⁴ The relationship of the HRF AUC and the steady-state response to step changes in neuronal activation stems from the fact that the integral of the impulse response of a linear, time-invariant system is equivalent to the system's steady-state response to a unit step function.

response followed by a steady-state response. (B) Timings of novelty events presented to a subject during a hypothetical event-related fMRI experiment. (C) Hypothetical brain activation map, where ROI 1 corresponds to a task relevant region and ROI 2 to a task irrelevant region. (D) Hypothetical BOLD responses to the novelty events measured in ROI 1 (top) and ROI 2 (bottom). (E) HRF estimates for ROI 1 (top) and ROI 2 (bottom) obtained using the hypothetical BOLD responses shown in (E) and the timings of the novelty events in (B).

Comment 2: Given the small size of the LC – could partial volume effects be a significant issue? The voxel size in the BOLD signal were large relative the size of the LC. It has been found that the volume of the LC is associated with extent of tau- and Abeta pathology (smaller volume associated with higher pathology) – see for example n doi: 10.3389/fnana.2017.00080, doi: 10.1016/j.brainres.2017.12.027 , doi: 10.3233/JAD-142445. Thus, a smaller LC may mean a lower activation level and thus be a confound with the computed associations.

Reply: We thank the reviewer for the question. We agree that the small size of the LC as well as its location close to the fourth ventricle makes imaging this nucleus susceptible to partial volume effects. To account for this, we employed stringent preprocessing techniques (weighted registration of the brainstem, motion correction, nuisance regression, special smoothing using an ellipsoid Gaussian kernel) aiming to minimize the contribution of non-neural effects in the LC-BOLD signal due to the proximity of the LC to the 4th ventricle (Lines 540-543). Also, to illustrate that the BOLD signal in the 4th ventricle and the LC is uncorrelated after preprocessing we calculated pairwise Pearson’s correlations for each participant and each functional run (Fig. R5 – Fig. S20 in the *Supplementary material*), which confirmed that our original findings are not biased by the proximity of the LC to the 4th ventricle. In addition, as mentioned in our reply to *Comment 1 from Reviewer 1 on page 1*, by using an ellipsoid kernel we are more likely to reduce partial volume effects than previous studies using spherical kernels. This ellipsoid smoothing kernel also renders our spatial resolution of the data after smoothing comparable, and in some cases better than several other recent and relevant LC studies in the literature that performed data smoothing using a spherical Gaussian kernel of 6 mm FWHM or greater, regardless of the original voxel size resolution (reply to *Comment 1 from Reviewer 1 on page 1*). Therefore, we believe that the confounding effects due to partial volume in our results are at least comparable to the reported results in other relevant studies in the literature. Also, to illustrate that our results are not biased from the smoothing applied to the data, and consequently from the partial volume effects associated with it, we repeated our voxel-wise analysis for detecting brain activation during novelty versus repetition using unsmoothed data. As shown in Fig. R2 on page 8 and Fig. R4 on page 8 of this response letter, the results obtained with the unsmoothed data are very similar to our original results shown in Fig. 3-4 in the manuscript. A relevant discussion on this issue has been added in the revised manuscript as follows (Lines 545 – 550).

In the Discussion section on page 24 (first paragraph – lines 545-550)

This ellipsoid smoothing brought the resolution of our data to a comparable and sometimes even better resolution than other LC studies using spherical smoothing (Supplementary results S1). Furthermore, our sensitivity analyses performed using unsmoothed data (providing a maximum spatial resolution), an eroded version of the LC ROI, and the Replication and Matched Datasets demonstrated the robustness and reproducibility of our findings.

With respect to atrophy or volume loss, autopsy studies as well as work from our own group has demonstrated that neurons of the LC do not die early in the disease process, despite the structural (tau) and functional changes it is undergoing^{55, 56}. To further investigate the effect of grey matter

volume, we repeated the same analysis using grey matter density as a voxel-wise covariate. Age and sex were also included in the model as covariates. As illustrated in Fig. R13 (Fig. S3 in the *Supplementary material*), the results are similar to our original findings shown in Fig. 3 in the manuscript, suggesting that our associations for LC function are not affected by potential grey matter difference observed in our cohort. This has been added in the *Results* section of the revised manuscript (Lines 200-203).

In the Results section on page 9 (second paragraph – lines 200 - 203)

Our sensitivity analyses reproduced these observations when the unsmoothed data (Fig. S1) and the Replication Dataset (Fig. S2) were used, as well as when grey matter density was included into the model as a covariate (Fig. S3).

Figure R13: Top panel: validation results for brain activation during novelty versus repetition using grey matter density as a covariate. Age and sex were also included in the model as covariates.

Bottom panel: Fig. 3 in the submitted manuscript illustrating the same result obtained with grey matter density being excluded.

Comment 3: The LC is the major source of noradrenaline (NA) in the brain and is widely diffused across the brain, and in particular, underpins arousal, alertness and attention – see for example 10.1523/JNEUROSCI.1164-19.2019, doi: 10.1155/2017/6031478, doi: 10.1038/nrn2573. A theory of cognitive reserve gives prominent role to the right hemisphere (doi: 10.1016/j.neurobiolaging.2013.11.028) and it is quite interesting that there seems that stronger associations were found with this hemisphere and with tasks that are known to be right hemisphere dominant. It may be that novelty contributes to increased arousal and attention relative the familiar face-name pairs.

Reply: We thank the reviewer for this comment. Indeed, the relation between our findings showing stronger associations between activity in the right LC and A β -related cognitive decline with the idea of cognitive reserve is intriguing. The right lateralization observed in our findings is consistent with the involvement of the LC in cognitive processes, such as novelty, arousal, and attention^{29, 43}. Indeed, examination of encoding of novel face-name pairs compared to viewing familiar face-name pairs has been shown to activate regions implicated in novelty detection⁵⁷. In addition, as suggested by Robertson (2014) and the work of others, these processes have been shown to activate predominantly the right fronto-parietal network⁴⁴, suggesting a prominent role of the right hemisphere in cognitive reserve³⁰.

Our findings indicate that individuals who are able to maintain optimal levels of novelty-related LC functional properties may be more resilient to cognitive decline, even in the presence of elevated A β . Animal and human imaging studies provided evidence that novelty-related activation of the LC leads to reconfiguration of task-specific networks, such as the salience or memory-related networks^{58, 59}. Hence, maintenance of novelty-related activation of the right LC in these individuals may be reflective of functional network (re)organization promoting resiliency against the extant pathology. A relevant discussion has been added in the revised manuscript (Lines 467 - 482).

In the Discussion section on page 21 (second paragraph – lines 467-482)

It is known that beyond the LC's effects on learning⁴⁵, its putative function involves fast disengagement from specific tasks and amplification of attentional focus to the goal-relevant information. Additionally, allocation of attention has generally been ascribed to the right hemisphere⁴⁶, which aligns with the abundant involvement of attention in the digit symbol substitution test and our observation of lower right-sided LC activity being associated with steeper A β -related cognitive decline. This right lateralization is also consistent with the hypothesis that the LC may represent an important biological substrate underlying cognitive reserve and its associated processes, such as arousal, attention and novelty^{29, 43}, which all have been related to activation of predominantly the right fronto-parietal network^{30, 44}.

Previous imaging work showed lower connectivity between the LC and frontoparietal or salience networks resulting in greater distractibility in older individuals^{47, 48}. Our results

indeed demonstrated that individuals who are able to maintain optimal levels of novelty-related LC functional properties may be more resilient to cognitive decline, even in the presence of elevated A β . Together, these findings strengthen the role of the LC-NE system in modulating networks, promoting cognition and potentially supporting cognitive reserve.

Comment 4: The results from figure 5(b) would seem to indicate that the decrease in PACC5 scores is quite slow (it seems about 0.5 SD in 7 years in most affected group) – would this level of decline be associated with conversion to MCI/AD. It is not clear from study if participants remained healthy status over the time frame of the study (within the time frame of each participant's follow up period). I am asking as the discussion makes reference to prodromal AD, and AD patients, yet as far as is reported, the participants are 'healthy' status over the time frame of the follow up period.

Reply: We thank the reviewer for this comment. The estimated average PACC5 change per year is -0.08 (SD/Year). For the most affected group (-1SD of LC activity), the estimated average PACC5 change per year is -0.11 (SD/Year). During the course of the study, ten of our participants progressed to MCI, and five of them progressed further to AD. It should be noted that the criterion for MCI diagnosis in HABS is based on a global CDR of 0.5. The CDR is completed by accredited neuropsychologists and psychiatrists. All CDR raters are independent and blinded to participant biomarker status. Diagnosis is based on clinical consensus after reviewing multiple consecutive CDR evaluations, cognitive data including the Logical Memory delayed recall score, MMSE and GDS, as well as relevant medical history. As illustrated in Fig. R14 (Fig. R21C in the *Supplementary material*), the participants who progressed to MCI/AD during the course of the study exhibit the steepest A β -related cognitive decline, suggesting that our findings are applicable to both preclinical and prodromal AD. However, the limited number of prodromal AD patients does not allow us to make any separate conclusions on the role of the LC for cognitive decline in these individuals. Following the reviewers suggestion the criteria for MCI/AD diagnosis and number of participants who progressed to MCI/AD have been added in the *Methods* section (Lines 602 - 614) and a relevant discussion in the *Discussion* section (Lines 440-444).

Figure R14: Spaghetti plots of individual PACC5 trajectories with respect to the baseline visit, color coded by consensus diagnosis (cognitively normal, mild cognitive impairment, or dementia). For each panel the average unadjusted slope is also plotted (black solid lines). Number of participants $n = 128$ and number of observations is 753.

In the Discussion section on page 20 (second paragraph – lines 440-444)

Interestingly, during the course of our study, ten participants progressed to MCI/AD, suggesting that our findings are also applicable to prodromal AD. Future studies with longer follow-up or a larger group of prodromal AD are needed to examine whether the relationship between LC function and $A\beta$ -related cognitive decline varies as a function of disease stage.

In the Methods section on page 26 (first paragraph – lines 602-614)

Clinical disease progression during the course of the study was determined based on a consensus diagnosis of MCI and AD dementia. The criterion for MCI diagnosis was based on a (i) global CDR of 0.5 or (ii) performance on any domain-specific composite score (memory, processing speed, and executive function) lower than 1.5 standard deviations below the sample mean⁶⁰. The CDR was completed by accredited neuropsychologists and psychiatrists. All CDR raters were independent and blinded to participant biomarker status. Participants meeting these criteria were brought to a consensus meeting conducted by a multidisciplinary team of clinicians. Diagnosis was based on clinical consensus after reviewing multiple consecutive CDR evaluations, cognitive data including the Logical Memory delayed-recall score, MMSE and Geriatric Depression Scale (GDS)⁶¹, as well as relevant medical history. Based on this evaluation, 10 participants of our main sample progressed to MCI and 5 of them progressed further to AD dementia over the 10-year period of the study.

In the Supplementary material (Fig. S21)

Spaghetti plots of individual PACC5 trajectories with respect to the baseline visit, color coded by magnitude of novelty-related LC activity, LC functional connectivity or clinical disease progression. The color bars indicate (A) baseline NvR LC activity, (B) baseline NvR LC-Left amygdala/hippocampus FC, or (C) consensus diagnosis (cognitively normal, mild cognitive impairment, or dementia). For each panel the average unadjusted slope is also plotted (black solid lines). Number of participants $n = 128$ and number of observations is 753.

Comment 5: I am surprised that study does not include CSF based measures of tau and amyloid - that would have one way to include measures of the relevant pathology over time.

Reply: We thank the reviewer for the question. Lumbar punctures are optional in HABS and we only have that information available for 21 participants at baseline and for 15 participants at year 2, which does not allow us to perform any baseline or longitudinal analysis due to limitations in statistical power.

Comment 6: Comparing figures 5(b) and 7(b), the left hand side xy plot, I would have expected that the y-axis would show the same range in PACC5 values – one figure it is about 0.5 SD, and the other 1SD. There is a consistency in the right-hand side plot between both figures.

Reply: We thank the reviewer for this comment. The estimated average PACC5 change per year is -0.08 (SD/Year). For the most affected group (-1SD of LC connectivity), the estimated average PACC5 change per year is -0.14 (SD/year). The scales of the y-axis for the line graphs with PACC5 have been made consistent between the two figures in the revised manuscript.

Comment 7: The authors used global measures of ABeta, would perhaps measures of ABeta in the specific ROI would have been a better measure of ABeta load? Particularly since the objective of the study was to demonstrate an association between Abeta and LC activity or functional connectivity to the MTL.

Reply: We thank the reviewer for this question. We agree that using local measures of A β could be an interesting research direction. However, we do worry that this may be too far from the scope of the current manuscript. In particular, because it is not yet clear how ‘early’ should be defined (as in: 1) earlier in time for a subject, or 2) lower in A β binding, or 3) both?). In addition, so far, there is not yet any consensus with regard to which regions make up these earliest regions of A β accumulation^{62, 63, 64, 65}. Determining these early regions also means taking into account the size of a region and determining noise levels (lower binding of PiB is more likely to contain nonspecific binding). Such an analysis would thus bring in multiple other research questions that would not fit within this manuscript. We believe that using a more global measure of A β is a reliable estimation of A β burden.

References

1. Keren NI, Lozar CT, Harris KC, Morgan PS, Eckert MA. In vivo mapping of the human locus coeruleus. *NeuroImage* **47**, 1261-1267 (2009).
2. Del Cerro I, *et al.* Disrupted functional connectivity of the locus coeruleus in healthy adults with parental history of Alzheimer's disease. *J Psychiatr Res* **123**, 81-88 (2020).
3. von der Gablentz J, Tempelmann C, Münte TF, Heldmann M. Performance monitoring and behavioral adaptation during task switching: an fMRI study. *Neuroscience* **285**, 227-235 (2015).
4. Köhler S, Bär KJ, Wagner G. Differential involvement of brainstem noradrenergic and midbrain dopaminergic nuclei in cognitive control. *Hum Brain Mapp* **37**, 2305-2318 (2016).
5. Diedrichsen J. A spatially unbiased atlas template of the human cerebellum. *Neuroimage* **33**, 127-138 (2006).
6. Grueschow M, *et al.* Real-world stress resilience is associated with the responsivity of the locus coeruleus. *Nat Commun* **12**, 2275 (2021).
7. Munn BR, Müller EJ, Wainstein G, Shine JM. The ascending arousal system shapes neural dynamics to mediate awareness of cognitive states. *Nat Commun* **12**, 6016 (2021).
8. Yebra M, *et al.* Action boosts episodic memory encoding in humans via engagement of a noradrenergic system. *Nat Commun* **10**, 3534 (2019).
9. Lee TH, *et al.* Arousal increases neural gain via the locus coeruleus-norepinephrine system in younger adults but not in older adults. *Nat Hum Behav* **2**, 356-366 (2018).
10. Yebra M, *et al.* Action boosts episodic memory encoding in humans via engagement of a noradrenergic system. *Nat Commun* **10**, 3534 (2019).
11. Friston KJ, Worsley KJ, Frackowiak RS, Mazziotta JC, Evans AC. Assessing the significance of focal activations using their spatial extent. *Human brain mapping* **1**, 210-220 (1994).
12. Worsley KJ, Evans AC, Marrett S, Neelin P. A three-dimensional statistical analysis for CBF activation studies in human brain. *Journal of Cerebral Blood Flow & Metabolism* **12**, 900-918 (1992).

13. Benjamini Y, Heller R. False discovery rates for spatial signals. *Journal of the American Statistical Association* **102**, 1272-1281 (2007).
14. Jansen WJ, *et al.* Prevalence of cerebral amyloid pathology in persons without dementia: a meta-analysis. *JAMA* **313**, 1924-1938 (2015).
15. Jenkinson M, Beckmann CF, Behrens TE, Woolrich MW, Smith SM. FSL. *Neuroimage* **62**, 782-790 (2012).
16. Pruim RHR, Mennes M, van Rooij D, Llera A, Buitelaar JK, Beckmann CF. ICA-AROMA: A robust ICA-based strategy for removing motion artifacts from fMRI data. *NeuroImage* **112**, 267-277 (2015).
17. Braak H, Tredici KD. The pathological process underlying Alzheimer's disease in individuals under thirty. *Acta Neuropathol* **121**, 171-181 (2011).
18. Weinshenker D. Long Road to Ruin: Noradrenergic Dysfunction in Neurodegenerative Disease. *Trends Neurosci* **41**, 211-223 (2018).
19. Muresan Z, Muresan V. Neuritic deposits of amyloid-beta peptide in a subpopulation of central nervous system-derived neuronal cells. *Mol Cell Biol* **26**, 4982-4997 (2006).
20. Muresan Z, Muresan V. Seeding neuritic plaques from the distance: a possible role for brainstem neurons in the development of Alzheimer's disease pathology. *Neurodegener Dis* **5**, 250-253 (2008).
21. Braak H, Thal DR, Ghebremedhin E, Tredici KD. Stages of the Pathologic Process in Alzheimer Disease: Age Categories From 1 to 100 Years. *J Neuropathol Exp Neurol* **70**, 960-969 (2011).
22. Kelly L, *et al.* Identification of intraneuronal amyloid beta oligomers in locus coeruleus neurons of Alzheimer's patients and their potential impact on inhibitory neurotransmitter receptors and neuronal excitability. *Neuropathol Appl Neurobiol* **47**, 488-505 (2021).
23. Palop JJ, *et al.* Aberrant excitatory neuronal activity and compensatory remodeling of inhibitory hippocampal circuits in mouse models of Alzheimer's disease. *Neuron* **55**, 697-711 (2007).

24. Busche MA, *et al.* Critical role of soluble amyloid-beta for early hippocampal hyperactivity in a mouse model of Alzheimer's disease. *Proc Natl Acad Sci U S A* **109**, 8740-8745 (2012).
25. Busche MA, *et al.* Tau impairs neural circuits, dominating amyloid- β effects, in Alzheimer models in vivo. *Nature neuroscience* **22**, 57-64 (2019).
26. Schultz AP, *et al.* Phases of Hyperconnectivity and Hypoconnectivity in the Default Mode and Salience Networks Track with Amyloid and Tau in Clinically Normal Individuals. *The Journal of neuroscience* **37**, 4323-4331 (2017).
27. Zhang F, *et al.* beta-amyloid redirects norepinephrine signaling to activate the pathogenic GSK3 beta/tau cascade. *Science translational medicine* **12**, eaay6931 (2020).
28. Mather M. Noradrenaline in the aging brain: Promoting cognitive reserve or accelerating Alzheimer's disease? *Semin Cell Dev Biol* **116**, 108-124 (2021).
29. Robertson IH. A noradrenergic theory of cognitive reserve: implications for Alzheimer's disease. *Neurobiol Aging* **34**, 298-308 (2013).
30. Robertson IH. Right hemisphere role in cognitive reserve. *Neurobiol Aging* **35**, 1375-1385 (2014).
31. Stern Y. Cognitive reserve. *Neuropsychologia* **47**, 2015-2028 (2009).
32. Stern Y. Cognitive reserve in ageing and Alzheimer's disease. *Lancet Neurol* **11**, 1006-1012 (2012).
33. Veyrac A, Sacquet J, Nguyen V, Marien M, Jourdan F, Didier A. Novelty determines the effects of olfactory enrichment on memory and neurogenesis through noradrenergic mechanisms. *Neuropsychopharmacology* **34**, 786-795 (2009).
34. Landau SM, *et al.* Association of lifetime cognitive engagement and low β -amyloid deposition. *Arch Neurol* **69**, 623-629 (2012).
35. Valenzuela MJ, *et al.* Multiple biological pathways link cognitive lifestyle to protection from dementia. *Biol Psychiatry* **71**, 783-791 (2012).
36. Mohammed AK, Jonsson G, Archer T. Selective lesioning of forebrain noradrenaline neurons at birth abolishes the improved maze learning performance induced by rearing in complex environment. *Brain Res* **398**, 6-10 (1986).

37. Li B, Chohan MO, Grundke-Iqbal I, Iqbal K. Disruption of microtubule network by Alzheimer abnormally hyperphosphorylated tau. *Acta Neuropathol* **113**, 501-511 (2007).
38. Clewett DV, Lee TH, Greening S, Ponzio A, Margalit E, Mather M. Neuromelanin marks the spot: identifying a locus coeruleus biomarker of cognitive reserve in healthy aging. *Neurobiol Aging* **37**, 117-126 (2016).
39. Laeng B, Ørbo M, Holmlund T, Miozzo M. Pupillary Stroop effects. *Cogn Process* **12**, 13-21 (2011).
40. Murphy PR, Robertson IH, Balsters JH, O'Connell R G. Pupillometry and P3 index the locus coeruleus-noradrenergic arousal function in humans. *Psychophysiology* **48**, 1532-1543 (2011).
41. Steiner GZ, Barry RJ. Pupillary responses and event-related potentials as indices of the orienting reflex. *Psychophysiology* **48**, 1648-1655 (2011).
42. Borodovitsyna O, Flamini M, Chandler D. Noradrenergic Modulation of Cognition in Health and Disease. *Neural Plast* **2017**, 6031478 (2017).
43. Sara SJ. The locus coeruleus and noradrenergic modulation of cognition. *Nat Rev Neurosci* **10**, 211-223 (2009).
44. Singh-Curry V, Husain M. The functional role of the inferior parietal lobe in the dorsal and ventral stream dichotomy. *Neuropsychologia* **47**, 1434-1448 (2009).
45. Jacobs H, *et al.* Dynamic behavior of the locus coeruleus during arousal-related memory processing in a multi-modal 7T fMRI paradigm. *Elife* **9**, (2020).
46. Mesulam MM. A cortical network for directed attention and unilateral neglect. *Ann Neurol* **10**, 309-325 (1981).
47. Lee T-H, *et al.* Arousal increases neural gain via the locus coeruleus–noradrenaline system in younger adults but not in older adults. *Nature Human Behaviour* **2**, 356-366 (2018).
48. Lee T-H, Kim SH, Katz B, Mather M. The decline in intrinsic connectivity between the salience network and locus coeruleus in older adults: implications for distractibility. *Frontiers in aging neuroscience* **12**, 2 (2020).

49. Poe GR, *et al.* Locus coeruleus: a new look at the blue spot. *Nat Rev Neurosci* **21**, 644-659 (2020).
50. Hansen N. The Longevity of Hippocampus-Dependent Memory Is Orchestrated by the Locus Coeruleus-Noradrenergic System. *Neural Plast* **2017**, 2727602 (2017).
51. Lustberg D, Iannitelli AF, Tillage RP, Pruitt M, Liles LC, Weinshenker D. Central norepinephrine transmission is required for stress-induced repetitive behavior in two rodent models of obsessive-compulsive disorder. *Psychopharmacology (Berl)* **237**, 1973-1987 (2020).
52. Lindquist MA, Meng Loh J, Atlas LY, Wager TD. Modeling the hemodynamic response function in fMRI: efficiency, bias and mis-modeling. *Neuroimage* **45**, S187-198 (2009).
53. Poublanc J, *et al.* Measuring cerebrovascular reactivity: the dynamic response to a step hypercapnic stimulus. *J Cereb Blood Flow Metab* **35**, 1746-1756 (2015).
54. Prokopiou PC, Pattinson KTS, Wise RG, Mitsis GD. Modeling of dynamic cerebrovascular reactivity to spontaneous and externally induced CO₂ fluctuations in the human brain using BOLD-fMRI. *Neuroimage* **186**, 533-548 (2019).
55. Jacobs HIL, *et al.* In vivo and neuropathology data support locus coeruleus integrity as indicator of Alzheimer's disease pathology and cognitive decline. *Sci Transl Med* **13**, eabj2511 (2021).
56. Theofilas P, Dunlop S, Heinsen H, Grinberg LT. Turning on the Light Within: Subcortical Nuclei of the Isodentritic Core and their Role in Alzheimer's Disease Pathogenesis. *J Alzheimers Dis* **46**, 17-34 (2015).
57. Sperling RA, Bates JF, Cocchiarella AJ, Schacter DL, Rosen BR, Albert MS. Encoding novel face-name associations: a functional MRI study. *Hum Brain Mapp* **14**, 129-139 (2001).
58. Tulving E, Kroll N. Novelty assessment in the brain and long-term memory encoding. *Psychon Bull Rev* **2**, 387-390 (1995).
59. Zerbi V, *et al.* Rapid reconfiguration of the functional connectome after chemogenetic locus coeruleus activation. *Neuron* **103**, 702-718. e705 (2019).
60. Orlovsky I, *et al.* The relationship between recall of recently versus remotely encoded famous faces and amyloidosis in clinically normal older adults. *Alzheimers Dement (Amst)* **10**, 121-129 (2018).

61. Yesavage JA, *et al.* Development and validation of a geriatric depression screening scale: a preliminary report. *J Psychiatr Res* **17**, 37-49 (1982).
62. Farrell ME, Chen X, Rundle MM, Chan MY, Wig GS, Park DC. Regional amyloid accumulation and cognitive decline in initially amyloid-negative adults. *Neurology* **91**, e1809-e1821 (2018).
63. Grothe MJ, Barthel H, Sepulcre J, Dyrba M, Sabri O, Teipel SJ. In vivo staging of regional amyloid deposition. *Neurology* **89**, 2031-2038 (2017).
64. Guo T, Landau SM, Jagust WJ. Detecting earlier stages of amyloid deposition using PET in cognitively normal elderly adults. *Neurology* **94**, e1512-e1524 (2020).
65. Palmqvist S, *et al.* Earliest accumulation of β -amyloid occurs within the default-mode network and concurrently affects brain connectivity. *Nat Commun* **8**, 1214 (2017).

Reviewers' Comments:

Reviewer #1:

Remarks to the Author:

Prokopiou, Engels-Dominguez et al have resubmitted a revised version of the manuscript "Lower novelty-related locus coeruleus function is associated with A β -related cognitive decline in clinically normal individuals". The manuscript has been in part rewritten and clarifications have been added substantially improving the manuscript. Importantly, a replication dataset has been added and the main findings of greater neuronal activity in the Locus Coeruleus and functional connectivity between LC and MTL on novelty versus repetition were replicated in this dataset.

However, it seems the cognitive results were not replicated in the replication dataset. At least it is not stated they were. This should be clearly stated in the results section.

I understand that it may be difficult finding a replication dataset large enough for replicating the correlations of LC activity and functional connections with cognitive decline and Ab.

This said, the results concerning lateralization and cognition thereby remain more uncertain and weaker. They need to be treated accordingly.

Please include a clear limitation statement in the limitation section that states that the results relating to cognitive change were not replicated in an independent cohort and that the results should be considered preliminary until replicated.

Minor:

Line 527 contains a repetition of the statement: "via molecular mechanisms such as β -adrenergic enhanced neurogenesis 59 and elevated expression of plasticity-related genes 60".

Reviewer #2:

Remarks to the Author:

The authors have addressed all of my concerns.

Reviewer #3:

Remarks to the Author:

The authors have thoroughly and adequately addressed my comments, and the manuscript is significantly strengthened by the revisions.

Reviewer #4:

Remarks to the Author:

The authors have responded to my comments with great care and detail.

The manuscript contributes to our understanding of the role of LC plays in the development of AD - the study investigated this issue in a cohort of older healthy subjects who were undergoing cognitive decline. The combination with the face-name pair fmri task (well validated), and analysis are a real strength of this study. The addition of a replication dataset further supports the robustness and reproducibility of the results.

Response letter: NCOMMS-21-27589A-Z

We would like to thank the editor and reviewers for their precious time on reviewing our manuscript, as well as for their thoughtful questions, comments and efforts that helped us improve clarity and presentation of our findings. Some remaining comments from Reviewer #1 have been addressed. Relevant changes related to these comments are copy-pasted (in green) along with their location in the manuscript in the text boxes below the comments.

Reviewer #1 (Remarks to the Author):

Prokopiou, Engels-Dominguez et al have resubmitted a revised version of the manuscript “Lower novelty-related locus coeruleus function is associated with A β -related cognitive decline in clinically normal individuals”. The manuscript has been in part rewritten and clarifications have been added substantially improving the manuscript. Importantly, a replication dataset has been added and the main findings of greater neuronal activity in the Locus Coeruleus and functional connectivity between LC and MTL on novelty versus repetition were replicated in this dataset. However, it seems the cognitive results were not replicated in the replication dataset. At least it is not stated they were. This should be clearly stated in the results section.

I understand that it may be difficult finding a replication dataset large enough for replicating the correlations of LC activity and functional connections with cognitive decline and Ab. This said, the results concerning lateralization and cognition thereby remain more uncertain and weaker. They need to be treated accordingly. Please include a clear limitation statement in the limitation section that states that the results relating to cognitive change were not replicated in an independent cohort and that the results should be considered preliminary until replicated.

Reply: We would like to thank the reviewer for the careful and insightful review of our manuscript. Regarding the replication of our cognitive results, we agree with the reviewer that finding a replication dataset large enough that would allow us to replicate our findings is challenging given the uniqueness of our dataset (baseline task-related fMRI and PiB-PET and longitudinal cognitive measurements over a period of 10 years). This remains to be performed in a future study. A relevant comment has been added in the limitations section.

*In the Discussion section on page 14 (second paragraph – lines 387-390)
Finding a replication dataset large enough that would allow us to perform similar sensitivity analyses for the association of LC function with A β -related cognitive decline is challenging and remains to be performed in a future study.*

Minor:

Line 527 contains a repetition of the statement: “via molecular mechanisms such as β -adrenergic enhanced neurogenesis 59 and elevated expression of plasticity-related genes 60”.

Reply: We thank the reviewer for the comment. The repetition statement has been removed.

*In the Discussion section on page 14 (first paragraph – lines 365-368)
Animal research has suggested that greater novelty-related LC activity may be a potentially important component mediating the cognitive effects promoting cognitive reserve 56 via molecular mechanisms such as β -adrenergic enhanced neurogenesis⁵⁷ and elevated expression of plasticity-related genes^{56, 58}.*

Reviewer #2 (Remarks to the Author):

The authors have addressed all of my concerns.

Reply: We would like to thank the reviewer for the careful and insightful review of our manuscript.

Reviewer #3 (Remarks to the Author):

The authors have thoroughly and adequately addressed my comments, and the manuscript is significantly strengthened by the revisions.

Reply: We would like to thank the reviewer for the careful and insightful review of our manuscript.

Reviewer #4 (Remarks to the Author):

The authors have responded to my comments with great care and detail. The manuscript contributes to our understanding of the role of LC plays in the development of AD - the study investigated this issue in a cohort of older healthy subjects who were undergoing cognitive decline. The combination with the face-name pair fmri task (well validated), and analysis are a real strength of this study. The addition of a replication dataset further supports the robustness and reproducibility of the results.

Reply: We would like to thank the reviewer for the careful and insightful review of our manuscript.